



# Afforestation impacts on terrestrial hydrology insignificant compared to climate change in Great Britain

Marcus Buechel[1], Louise Slater[1], Simon Dadson[1,2]

[1]School of Geography and the Environment, University of Oxford, South Parks Road, Oxford OX1 3QY, UK.
[2]UK Centre for Ecology & Hydrology, Crowmarsh Gifford, Wallingford, OX10 8BB, UK.

*Correspondence to*: Marcus Buechel (marcus.buechel@ouce.ox.ac.uk)

**Abstract.** Widespread afforestation has been proposed internationally to reduce atmospheric carbon dioxide, however the specific hydrological consequences and benefits of such large-scale afforestation (e.g., Natural Flood Management) are poorly understood. We use a high-resolution land surface model, JULES, with realistic potential afforestation
scenarios to quantify possible hydrological change across Great Britain in both present and projected climate. We assess whether proposed afforestation produces significantly different regional responses across regions; whether hydrological fluxes, stores and events are significantly altered by afforestation relative to climate; and how future hydrological processes may be altered up to 2050. Additionally, this enables determination of the relative sensitivity of land surface process representation in JULES compared to climate changes. For these three aims we run simulations using: (i) past
climate with proposed land cover changes and known floods and drought events; (ii) past climate with independent changes in precipitation, temperature, and $CO_2$; and (iii) a potential future climate (2020-2050). We find the proposed scale of afforestation is unlikely to significantly alter regional hydrology, however it can noticeably decrease low flows whilst not reducing high flows. The afforestation levels minimally impact hydrological processes compared to changes in precipitation, temperature, and $CO_2$. Warming average temperatures (+ 3 °C) decreases streamflow, while rising
precipitation (130%) and $CO_2$ (600 ppm) increase streamflow. Changes in high flow are generated because of evaporative parameterisations whereas low flows are controlled by runoff model parameterisations. In this study, land surface parameters within a land surface model do not substantially alter hydrological processes when compared to climate.

## 1 Introduction

Land cover (e.g., grassland and bare ground) exerts a strong control on catchment hydrology (Blöschl *et al.*, 2007; Pattison
and Lane, 2012; Rogger *et al.*, 2017). A land use or land cover change (LULC) can obscure the impact of climate on streamflow. LULC alters streamflow by changing hydrological processes (e.g., subsurface flow) over multiple spatial and temporal scales. One example of LULC is afforestation, which can lower catchment water tables (Kellner and Hubbart, 2018; Shuttleworth *et al.*, 2019; Peskett *et al.*, 2020), increase precipitation downwind (Teuling *et al.*, 2019; Meier *et al.*, 2021; Xu *et al.*, 2022) and alter transpiration rates over time (Newson and Calder, 1989; Hudson *et al.*, 1997; Marc and
Robinson, 2007). Many studies suggest afforestation can reduce overall streamflow, however, land management practices (e.g., artificial ditching and road cutting) may also increase peak streamflow (Beschta *et al.*, 2000; Bathurst *et al.*, 2018). It is therefore important to understand all the hydrological processes of afforestation (and other LULC), especially as the climate changes and the hydrological cycle intensifies (Kundzewicz, 2011; IPPC, 2019; Hung *et al.*, 2020). Changing rainfall (Gao *et al.*, 2018; Fowler *et al.*, 2021), temperature (Wasko, 2021) and carbon dioxide (Gedney *et al.*, 2006; Betts
*et al.*, 2007) could all influence how large-scale afforestation impacts hydrology.

Potential afforestation benefits include the reduction of atmospheric $CO_2$ (Griscom *et al.*, 2017; Hawes, 2018; Cook-Patton *et al.*, 2020; Palmer, 2021); moderation of temperature extremes (O'Briain *et al.*, 2020; Schwaab *et al.*, 2020);


reduction of air and noise pollution (Oldfield *et al.*, 2013; Fenner, 2017); provision of areas of societal wellbeing (Dick *et al.*, 2019); and reduction of flood risk as a form of Natural Flood Management (NFM) (Dadson *et al.*, 2017; Cooper *et al.*, 2021). These potential benefits have led governments and businesses to pledge woodland acreage increases (Lewis *et al.*, 2019; Seddon *et al.*, 2020), but there is a need to quantify the actual (and not merely perceived) advantages of planting the right trees in the right place (Grassi *et al.*, 2017; Seddon *et al.*, 2021). The UK government plans to expand woodland from 13% to 17% of land cover by 2050 to reach Net Zero (Committee on Climate Change, 2019b, 2019a). As the UK experiences more frequent and larger floods (Griffin *et al.*, 2019; Hannaford *et al.*, 2021), it is important to learn how afforestation could reduce flood peaks. Therefore, additional research is required to interrogate the hydrological response to afforestation over regional to continental scales.

Determining woodland influence on hydrology is not new (Andréassian, 2004). Studies investigating afforestation impact on streamflow have used global streamflow datasets (Bradshaw *et al.*, 2007; Do *et al.*, 2017), paired catchments (Bosch and Hewlett, 1982; Brown *et al.*, 2013; Bathurst *et al.*, 2018) and modelling (Iacob *et al.*, 2017; Li *et al.*, 2018; Speich *et al.*, 2018). Recent UK studies show how wet canopy evaporation can reduce high runoff (Page *et al.*, 2020) and afforestation can increase saturated soil hydraulic conductivity in upland areas (Murphy *et al.*, 2021). These results, and others, have been taken by some to suggest that afforestation can reduce flooding with greater water usage, higher infiltration rates and increased floodplain hydraulic roughness (Nisbet *et al.*, 2011; Cooper *et al.*, 2021). However, it remains unclear whether hydrological afforestation effects have a detectable hydrological impact over areas larger than 50 km$^2$ due to the small scales of existing studies of afforestation hydrology (Dadson *et al.*, 2017; Rogger *et al.*, 2017; Nisbet and Thomas, 2021). Further uncertainty exists between empirical studies and model results regarding peak streamflow generation following afforestation (Stratford *et al.*, 2017; Carrick *et al.*, 2019), making it difficult to transfer afforestation impacts on catchment hydrology over large regions.

Process-based numerical models provide one way to understand the consequences of afforestation (Gush *et al.*, 2002; Bonan, 2008). They incorporate known processes to determine system responses to scenarios (e.g., projected climate). Many studies have applied numerical models to determine afforestation's role in reducing flood risk, although few have considered areas larger than a single catchment (Stratford *et al.*, 2017). Land surface models (LSMs) incorporate a suite of known Earth system processes and have been used to study countrywide and continental hydrology (Prudhomme *et al.*, 2012; Blyth *et al.*, 2021). Inclusion of plant functional types, nutrient cycling, physiological forcing and surface energy fluxes mean LSMs are well suited to investigate afforestation mechanistic impacts on the hydrosphere compared to simpler models. This is essential to assess the hydrological response to a system as complex as vegetation change (Rogger *et al.*, 2017). LSMs should therefore quantify projected hydrological changes whilst modellers determine if outputs are realistic. Previous work using an LSM has shown tree planting location has a minimal impact compared to the extent planted within catchments (Buechel *et al.*, 2022). It is unknown whether this finding is true across all regions of Great Britain in a changing climate and the fidelity of the model parameterisations.

In this study, the impact of increasing broadleaf afforestation on Great Britain's hydrology is analysed with a high-resolution LSM. The approach allows direct comparison of streamflow with and without afforestation on a country-wide scale, whilst determining processes and catchment attributes driving modelled hydrological changes. Three central questions are investigated about realistic afforestation scenarios influencing country-wide hydrology:

1. What is the hydrological response to afforestation across Great Britain and how does this vary regionally?





2. How much of an impact would realistic afforestation scenarios have on hydrological processes compared to potential changes in climate?

3. How might realistic afforestation alter future hydrological scenarios up to 2050?

We evaluate these questions by quantifying how plausible afforestation across Great Britain could influence catchment hydrology, identifying changes across the streamflow spectrum, and testing the fitness for purpose of model process representation.

## 2 Methods

### 2.1 Plausible Afforestation Scenarios

Several afforestation scenarios have been used to investigate the hydrological consequences of afforestation, such as the 'global restoration potential' dataset (Bastin *et al.*, 2019; Meier *et al.*, 2021; Hoek van Dijke *et al.*, 2022) but its realism has been questioned (Friedlingstein *et al.*, 2019; Wilkes *et al.*, 2020). Hydrological conclusions derived from such unlikely scenarios are therefore equally questionable. The UK Government's Net Zero strategy includes planting 30 000 hectares of trees, equivalent to sequestering 14 MtCO$_2$e, a year from 2024 onwards (Committee on Climate Change, 2018). This roughly approximates to 900 000 hectares of additional woodland across the country by 2050. Plausible afforestation scenarios for Great Britain should emphasis planting trees in areas that accomplish maximal societal benefits (Bradfer-Lawrence *et al.*, 2014; Burke *et al.*, 2021). This study produces credible afforestation extents with multiple purposes and minimal resistance [Figure 1] by altering three previously developed afforestation scenarios for the countries of Great Britain: England, Wales, and Scotland. Northern Ireland is not included as model driving datasets do not cover this region. The scenario for England is the Environment Agency's Working with Natural Processes program, which aims to reduce flood risk and restore the natural regulating function of catchments (Environment Agency, 2018). In Wales the Glastir Woodland Creation opportunities map is utilised that aims to plant trees for maximal benefits (Welsh Government, 2021). Finally for Scotland, the Woodland Expansion Advisory Group's map is used that identified areas with the greatest potential for woodland expansion (Sing and Aitkenhead, 2020).

Woodland extent is spatially constrained for the chosen afforestation scenarios, however further limits are applied to promote afforestation in low-risk locations. Principles to expand woodland into areas that minimally interfere with existing land practices are used, developed by the Scottish Woodland Expansion Advisory Group and the Forestry Commission (Sing and Aitkenhead, 2020). Afforestation takes place in: acid grassland, arable and horticultural areas, heather, heather grassland, improved grassland and neutral grassland as defined by the CEH Land Cover 2000 (Fuller *et al.*, 2002) [Figure 1]. Woodland expansion does not encroach on urban areas, existing woodland, shrubland, bare ground, inland water, or upon biodiversity-rich grasslands (by excluding it from priority habitat areas). To note, grassland afforestation can decrease soil carbon and not sequester additional carbon over the short term (Don *et al.*, 2009). Other spatial constraints for woodland expansion are summarised in Figure 1. Across Great Britain, this afforestation criteria creates a geographical imbalance with largest potential afforestation areas in Scotland followed by Wales then England [Figure 2]. This is due to more land being initially identified in Scotland and Wales for afforestation and greater areal constraints to afforestation in England. A maximal potential afforestation area of 3.53 million hectares is generated in Great Britain using these afforestation constraints (approximately 700 000 hectares less than the 'global restoration potential').





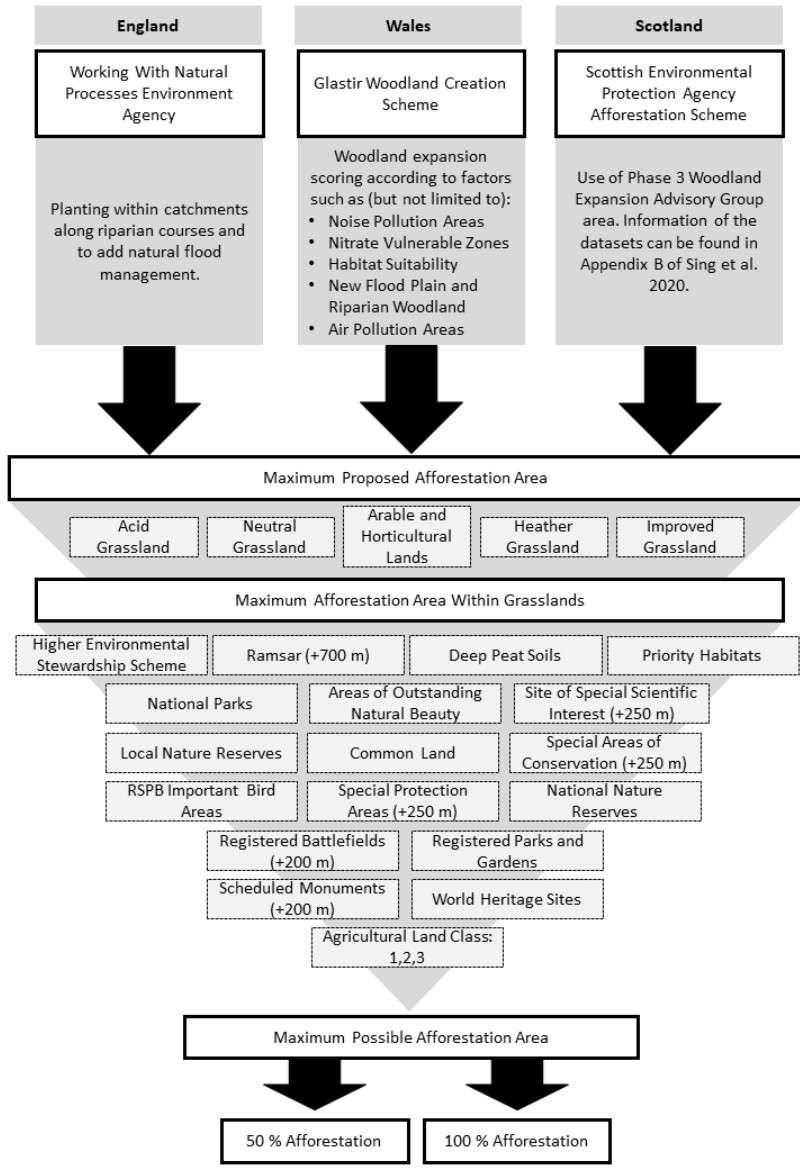

**Figure 1: Flow diagram explaining the creation of the two realistic afforestation scenarios. The top row indicates the three afforestation scenarios developed for England, Scotland, and Wales. The spatial extent of these scenarios is reduced by selecting the scenario areas that intersect with grasslands (as defined by the CEH Landcover 2000 map) and several other factors as shown above (such as Areas of Outstanding Natural Beauty and Special Protection Areas). 100 % and 50 % scenarios are created from the maximum possible afforestation area calculated.**

900 000 hectares of broadleaf woodland is randomly 'planted', at a 25 m resolution, within the generated maximum afforestation extent (on the same projected coordinate system as the CEH 2000 landcover map (Fuller *et al.*, 2002)). In addition, another scenario with approximately 450 000 hectares of woodland is made to represent afforestation at similar present rates (Forest Research, 2021a). Woodland extent across Great Britain changes from 12.3% to 14.3% and 16.2%



for the 50% and 100% afforestation scenarios respectively (using the CEH 2000 land cover). These two afforestation

scenarios are combined with the CEH 2000 landcover map (Fuller *et al.*, 2002) and scaled to a 1 km$^2$ grid, similar to the

CHESS-land dataset (Martinez-de la Torre *et al.*, 2018), by calculating the fraction of eight different land cover types

detailed in the following section. Arguably these scenarios are a restrictive level of afforestation, but they appear

ambitious when compared to current afforestation rates of approximately 10 000 hectares per year (Forest Research,

2021b).

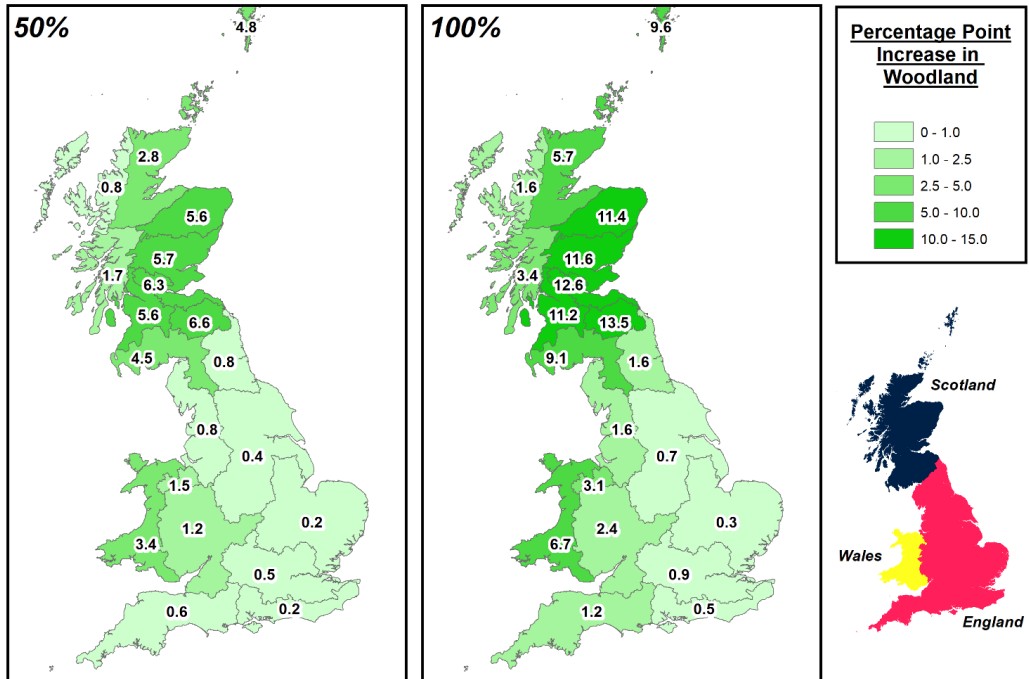

**Figure 2: Percentage point increase in broadleaf woodland for the two realistic afforestation scenarios generated for each of the twenty UKCP18 hydro-regions in Great Britain.**


### 2.2 Modelling Methodology

This study is split into three parts for each research question. This enables a coherent understanding of afforestation on

potential past, present and future hydrology within the model domain. First, the potential hydrological benefits, and

drawbacks, of afforestation scenarios are considered under present climate conditions. Second, a simple factorial

sensitivity analysis is undertaken to ascertain the relative importance of afforestation and climate on future hydrological

changes, and model process parameterisation. Finally, a future climate scenario is utilised to project how plausible future

afforestation may alter hydrology in Great Britain up to 2050.

### 2.2.1 Model Description

The Joint UK Land Environment Simulator (JULES), a physically based LSM, is used and simulates water, carbon and

energy stores and fluxes (Best *et al.*, 2011; Clark *et al.*, 2011). JULES has been used many times previously, for example:

determining the climatic impact of the Montreal protocol (Young *et al.*, 2021); assessing trends in evapotranspiration

across Great Britain since the 1960s (Blyth *et al.*, 2019); and aiding production of high-resolution UK soil moisture



datasets (Peng *et al.*, 2021). Due to the complexity and high number of free and fixed parameters of JULES the base validated model configuration of Buechel *et al.* (2022), Robinson *et al.* (2017) and Martínez-de la Torre *et al.* (2019) is utilised. Configuration details are found in Buechel *et al.* (2022) and accessible as Rose suite u-ce663 from the Met Office Rose/Cylc suite control system (https://metomi.github.io/rose/doc/html/index.html). The CHESS-land dataset specifies
JULES' bounding conditions including soil hydraulic, thermal, vegetation and orographic properties at a 1 km$^2$ spatial resolution (Martinez-de la Torre *et al.*, 2018). CHESS-met provides the meteorological information of air temperature, pressure, specific humidity as well as short- and long-wave radiation for the first two parts of the study (Robinson *et al.*, 2017). In the final study section, we utilise CHESS-SCAPE (Robinson *et al.*, 2022), a 1 km$^2$ downscaled UKCP18 projection (Lowe *et al.*, 2018) with all the same variables and spatial and temporal resolution of CHESS-met.

JULES runs at a numerical timestep of 30 minutes. The land surface is divided into eight possible types, five vegetated (broadleaf, needleleaf, C3 grass, C4 grass (crops), shrubs) and three non-vegetated (urban, inland water, bare soil). Precipitation covers a grid cell according to a constant dependent on temperature (Best *et al.*, 2009) and is intercepted by vegetation, as a function of the leaf area index (LAI). Canopy throughfall is a function of the existing canopy water, rainfall, and the maximum amount of canopy water (related to LAI). Evapotranspiration is calculated using effective
surface resistance (water stored in the canopy compared to the maximum canopy capacity) and stomatal conductance is modelled using soil moisture, atmospheric carbon dioxide and the vapour pressure deficit (Cox *et al.*, 1998). Water at the surface is routed as infiltration excess overland flow, at a rate controlled by the soil hydraulic conductivity, or as saturation-excess overland flow calculated by the Probability Distributed Model (PDM) (Moore, 2007; Clark and Gedney, 2008). A topography-derived parametrisation of soil water storage in the PDM is used to calculate grid fraction saturation,
which is used as a multiplier to convert excess water reaching the surface to saturation-excess overland flow (Martínez-De La Torre *et al.*, 2019; Lewis and Dadson, 2021). Water flux through the four soil layers (3 m deep) is calculated by the Darcy-Richards equation and the van Genuchten scheme calculates suction and soil conductivity (van Genuchten, 1980). Vegetation extracts excess water from the different soil layers as a function of root density and soil moisture critical and wilting points. As soil layers become progressively saturated water is passed downwards until excess water at the
base becomes subsurface runoff. The River Flow Model (RFM) implementation of the kinematic wave equation solution routes both the surface and subsurface runoff according to a D8 flow direction grid (Davies *et al.*, 2022). The model is spun-up for ten years in all experiments so that soil moisture content equilibrates (Martinez-de la Torre et al., 2018; Blyth et al., 2019a).

Model output (soil moisture, evaporation and streamflow) is validated with twelve COSMOS-UK stations (Cooper *et al.*, 2021) and the National River Flow Archive database for the investigated catchments (Vitolo *et al.*, 2016) [Supplementary Material: Figures S1 & S2]. The Kling-Gupta Efficiency Measure (KGE) (Gupta *et al.*, 2009) quantifies model accuracy where scores better than -0.41 show model performance greater than the mean seasonal cycle (Knoben *et al.*, 2019).
Several model outputs are validated to determine whether JULES is providing the right result for the right reason across model parameterisation domains (Mai et al., 2020; Lane et al., 2021). We find JULES performs satisfactorily: streamflow has a median KGE of 0.50 (minimum: 0.08; maximum: 0.76), soil moisture 0.44 (minimum: 0.20; maximum: 0.82) and potential evaporation 0.53 (minimum: 0.22; maximum: 0.72). These are not perfect scores and so caution must be applied when considering results. KGE scores illustrate the challenges in producing a 'model of everywhere' when there is so
much uncertainty in model parameterisations and parameters (Beven, 2007; Blair *et al.*, 2019). Furthermore, any changes in model results are relative to simulations and are not compared to realistic absolute values due to limitations of model





process representation. Refer to Buechel *et al.* (2022) for further details on model validation and Supplementary Material [Tables S1,S2,S3].

**2.2.2 Streamflow Analysis**

Streamflow is the combined output of upstream hydrological processes and thus invaluable to explore afforestation effects. Fifty-one catchments are selected to assess streamflow across all regions [Supplementary Material: Figure S1 & Table S4]. Ten are from the previous work of Buechel *et al.* (2022), Crooks *et al.* (2014), and Martínez-De La Torre et al. (2019), thirty-nine are from the UKBN2 gauging station network (Harrigan *et al.*, 2018), and two of the largest catchments in the Dee region (as there were no rivers fitting our criteria). UKBN2 catchments are near natural with

minimal human interference; important for isolating changes due to LULC and climate with minimal anthropogenic interference (Villarini and Wasko, 2021), which may be unrepresented within JULES. Chosen catchments are larger than 150 km$^2$ in size so that processes simulated are more faithful at JULES' spatial (1 km$^2$) and temporal (30 minute) resolution, apart from two rivers in Scotland which were needed so that all regions would have streamflow simulations. Chosen catchments cover all UKCP18 regions to enable analysis of how countrywide drought and flood conditions change

with afforestation.

Flow percentiles at the 1, 5, 10, 50, 90, 95, 99 % are calculated to observe flow change across the whole flow spectrum with afforestation over the full hydrological year and not just individual events. The slope of the flow duration curve (FDC) is calculated to understand flow variability changes:

$$FDC = \frac{\ln(Q_{33\%}) - \ln(Q_{66\%})}{(0.66 - 0.33)}$$
(1)

Where $Q_{33\%}$ and $Q_{66\%}$ are the 33$^{rd}$ and 66$^{th}$ percentile of streamflow respectively. Chosen streamflow metrics enable quantification of extreme and average changes induced by afforestation to learn how flow regimes could changes at present and the future.

**2.3 Present Hydrological Response to Afforestation**

The period of 2000-2015 is chosen for the first study part. This is a flood-rich period, including several drought events, allowing determination of the role afforestation would have on these events across the country (Wilby and Quinn, 2013; Dadson *et al.*, 2017). Hydrological processes are disaggregated by the twenty UKCP18 river basin boundaries (Lowe *et al.*, 2018) [Figure 2], which are hydrologically distinct to compare afforestation influence across Great Britain. Water flux and store changes (evaporation, soil moisture and runoff) are calculated seasonally and the whole period for each

region. The Theil-Sen slope estimator (Sen, 1968; Theil, 1992) is employed to obtain the percentage change in the hydrological variable relative to the percentage point increase in woodland for each region. For example, a catchment where its woodland area increased from 10% to 17% of its overall area would represent a 7% percentage point increase. This allows the sensitivity and relative hydrological response to afforestation to be quantified regarding the spatial scales of afforestation and catchments considered. Theil-Sen, a form of nonparametric regression, is more robust than ordinary

least squares regression as it is less sensitive to outliers (Helsel *et al.*, 2020), and it is frequently used in other studies (Gudmundsson *et al.*, 2019; Griffin *et al.*, 2022). The Theil-Sen slope estimators quantify both the direction and size of the response in water stores, fluxes and metrics to afforestation. Spearman's rank correlation coefficient (ρ) is used to identify the strength of the association between afforestation and different hydrological fluxes across regions, where each data point represents an afforestation scenario of a region. The values of ρ reveals the extent to which afforestation, or





factors other than afforestation (e.g. regional effects such as soil properties) influences the hydrology. For example, a weak Spearman's rank correlation (e.g., between 0.4 and -0.4) indicates that afforestation is not strongly associated with hydrological change, implying that regional effects are more important than the level of woodland planted.

**2.4 Proportional Influence of Afforestation Compared to Climate**

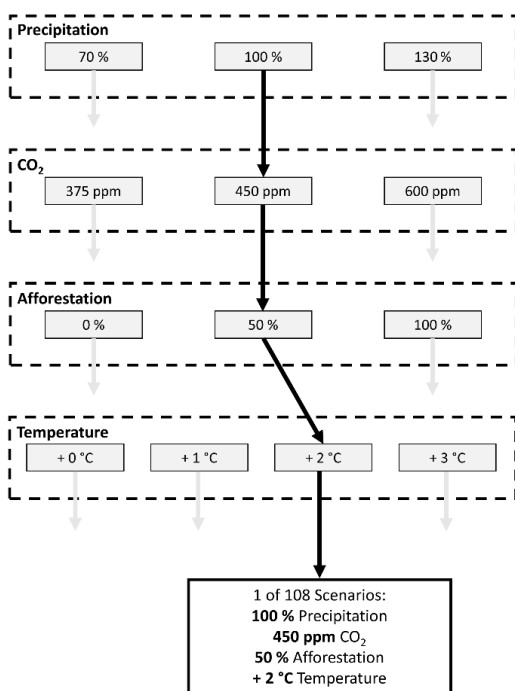


**Figure 3: Flowchart illustrating the potential scenarios generated with differences in precipitation, CO₂, afforestation, and temperature. An indicated pathway of one of the 108 scenarios is shown with the thick black arrows. Not all the scenarios seen here are possible or likely. A scenario of 0% afforestation, 100 % precipitation, 375 ppm and + 0 °C temperature would take us to a situation like the start of the 21st Century. A scenario of 0 % afforestation, 130 % precipitation, 600 ppm and + 3 °C**
**temperature would lead us to a situation of SSP5 ('business as usual' or extreme emissions scenario).**

JULES' hydrological sensitivity to potential future atmospheric and afforestation changes is determined by undertaking a factorial sensitivity analysis. This allows the terrestrial hydrological response to afforestation, within the model domain, to be verified relative to atmospheric drivers. Three variables are independently altered in the base meteorological driving
data (CHESS-met) for the period of 2000-2015: precipitation, temperature, and carbon dioxide. The model is spun-up for ten years (using 2000-2001) with the perturbed meteorological parameters. Maximal projected changes in precipitation and temperature, as stated in the UKCP18 scenarios, are the baseline for changing the meteorological data (Lowe *et al.*, 2018). In this way, atmospheric variables are altered within a range deemed physically plausible by a validated climate model to observe maximal sensitivity within a credible realm. From the original CHESS-met data, precipitation (in mm
day[-1]) is altered by 70 %, 100 % and 130 %; temperature is raised by 1 °C, 2 °C and 3 °C; atmospheric carbon dioxide is enhanced from 375 ppm to 450 ppm and 600 ppm. For temperature and precipitation, these are the approximate maximal changes in the UKCP18 scenarios up to 2050 under the RCP 8.5 scenario for the 95[th] percentile of the UKCP18





probabilistic projections (1981-2000 compared to 2041-2060). Current carbon dioxide levels are approximately 415 ppm, up from 375 ppm at the start of the 21st Century (NOAA, 2022). Both Gedney et al. (2006) and Blyth et al. (2019) have

explored the atmospheric carbon dioxide impacts on hydrology using JULES (or its predecessor) before, however this approach goes further to see how it relates to potential future climate and LULC. An ensemble of 108 different scenarios per region is generated [Figure 3], although this produces the full set of potential scenarios including unlikely ones. For example, an increase in carbon dioxide emissions with a decrease in both temperature and precipitation. Including the whole range not only accounts for the response to extremes in certain seasons, but also provides enough data to decompose

the contribution of each driver to hydrological change. This form of factorial sensitivity analysis was also undertaken due to the nonlinearity of hydrological process nonlinearity.

**2.5 Hydrological Response to Potential Future Climate**

The CHESS-SCAPE dataset is used to study future climate impacts (Robinson *et al.*, 2022). CHESS-SCAPE is a 1 km

resolution dataset downscaled from the 12 km UKCP18 climate projections (Met Office, 2018) used previously to investigate the influence of climate change on future UK hydrology (Kay, 2021; Griffin *et al.*, 2022). The 12 km simulations were generated using a perturbed parameter ensemble of the Met Office Hadley Centre Global Climate Model (HadGEM3-GA705) under the RCP 8.5 scenario (Murphy *et al.*, 2019) which is nested and forced by the wider 60 km global simulations. The CHESS-SCAPE dataset is created by selecting four model ensembles spanning the range of the

original 12 model perturbed parameter ensemble of UKCP18, and then downscaled according to the CHESS-met methodology using local topography (Robinson *et al.*, 2022). The RCP 8.5 uncorrected meteorological dataset for the period 2020-2050 forces JULES with the land cover scenarios across the 20 regions to enable identification of tree planting effect on modulating climatic extremes produced by the worst-possible case of $CO_2$ concentration in the atmosphere. RCP 8.5 is increasingly recognised as an unlikely and extreme scenario (Hausfather and Peters, 2020),

however it is useful for detecting the climate signal from the influence of afforestation and the only scenario available in UKCP18. Vegetation is fixed with no dynamic competition between the different vegetation types in our model setup of JULES and so vegetation does not need to be recalibrated to this new climate regime. The model is spun-up for the period 2010 to 2020 so that the hydrological system is in equilibrium.

**3 Results**

**3.1 Changes in Regional Hydrology with Afforestation**

Broadleaf afforestation across Great Britain clearly changes modelled evaporative processes [Figure 4] [Table 1]. Canopy and soil evaporation increase on average for the entire year which decreases canopy and soil water stores; however, the direction of change varies seasonally [Figure 4]. Averaged over the entire study period, overall canopy evaporation rises by 0.40 % (0.40 mm yr$^{-1}$) per percentage point of afforestation (PPPoA) with a moderate influence of location ($\rho = 0.59$)

and is greater in winter and less in summer months for almost all regions [Figure 4]. To reiterate, weaker Spearman rank correlation coefficients indicate factors other than afforestation extent are causing the variation in the hydrological response to afforestation. Canopy storage decreases by 0.73 % (0.001 mm) PPPoA and is minimally affected by afforestation location ($\rho = -0.94$). Simulated soil evaporation, including both evaporation from the soil surface and plant transpiration, increases with afforestation by 0.26 % (0.54 mm yr$^{-1}$) PPPoA and is partially dependent on geographic

region ($\rho = 0.53$). Soil evaporation is projected to increase substantially during winter and decrease in summer, particularly in Scottish regions. By contrast, modelled transpiration decreases with afforestation consistently throughout





the period (-0.59 % PPPoA) regardless of location (ρ = -0.97). Stomatal conductance decreases in summer and increases in winter and whilst it is sensitive to location no systematic pattern could be discerned (ρ = -0.021, p < 0.01). Notable variation is seen in the representation of stomatal conductance amongst LSMs and so this response could be particular to

the configuration used here.

**Table 1: Changes in the average water fluxes and stores with afforestation across Great Britain for each percentage point increase in broadleaf woodland for both the present climate and potential future climate. ρ (Spearman) correlations indicate the strength of association between increased afforestation and changes in the flux and stores where each data point represents**

**an afforestation scenario in a region. High absolute values (e.g., above 0.7) indicate planting location has a minimal influence on altering the response to afforestation. Values in bold are greater than an absolute value of 0.7.**

| | | Present (For each percentage point increase in woodland) | | | Future (For each percentage point increase in woodland) | | |
|---|---|---|---|---|---|---|---|
| | | Percentage Change (%) | Absolute Change (mm yr$^{-1}$ and mm) | ρ Correlation | Percentage Change (%) | Absolute Change (mm yr$^{-1}$ and mm) | ρ Correlation |
| Flux (mm yr$^{-1}$) | Canopy Evaporation | 0.40 | 0.40 | 0.59 | 0.33 | 0.47 | 0.66 |
| | Soil Evaporation | 0.26 | 0.54 | 0.53 | 0.11 | 0.29 | 0.51 |
| | Runoff | -0.30 | -1.9 | -0.85 | -0.27 | -1.84 | -0.74 |
| | Surface Runoff | -0.20 | -0.27 | -0.73 | -0.16 | -0.35 | -0.53 |
| | Subsurface Runoff | -0.34 | -1.0 | -0.85 | -0.26 | -0.95 | -0.78 |
| | Throughfall | -0.34 | -1.0 | -0.87 | -0.33 | -1.23 | -0.83 |
| Store (mm) | Total Soil Moisture | -0.05 | -0.4 | -0.83 | -0.047 | -0.41 | -0.77 |
| | Canopy Storage | -0.73 | -0.001 | -0.94 | -0.64 | -0.001 | -0.94 |


**Table 2: Changes in flow metrics with afforestation across Great Britain for each percentage point increase in broadleaf woodland for both the present climate and potential future climate. ρ (Spearman) correlations indicate the strength of association between increased afforestation and changes in the afforestation extent.**

| Flow Metric | Present (For each percentage point increase in woodland) | | Future (For each percentage point increase in woodland) | |
|---|---|---|---|---|
| | Percentage Change (%) | ρ Correlation | Percentage Change (%) | ρ Correlation |
| Very High (1%) | -0.054 | -0.2 | -0.11 | -0.44 |
| High (5%) | -0.11 | -0.54 | -0.11 | -0.66 |
| High (10%) | -0.11 | -0.6 | -0.09 | -0.63 |
| Median (50%) | -0.18 | -0.65 | -0.14 | -0.68 |
| Low (90%) | -0.24 | -0.66 | -0.25 | -0.64 |
| Lower (95%) | -0.32 | -0.71 | -0.34 | -0.75 |
| Very Low (99%) | -0.57 | -0.81 | -0.72 | -0.84 |
| Duration Curve | 0.09 | -0.2 | 0.053 | -0.37 |



Afforestation across Great Britain moderately reduces average river flow by 0.17 % PPPoA over the year with only a slight locational variation (ρ = -0.9). This decline in river flow is caused by decreasing surface and subsurface runoff (-0.20 %; -0.27 mm yr$^{-1}$ and -0.34 %; -1.03 PPPoA respectively) [Table 1]. Despite the consistent reduction in runoff components throughout the year, the response varies minimally by region (surface: ρ = -0.73, subsurface: ρ = -0.85) [Table 1]. Canopy throughfall, the simulated source of water for runoff, decreases with afforestation (-0.34 %; -1.00 mm yr$^{-1}$ PPPoA) regardless of planting location (ρ = -0.87) [Table 1], and its reduction is consistent throughout the year. Whilst average total soil moisture decreases with afforestation minimally (0.046 %, -0.40 mm PPPoA) and without influence of planting location (ρ = -0.83) [Table 1], the moisture available to vegetation from the uppermost soil layer is noticeably influenced by planting location (ρ = -0.59).

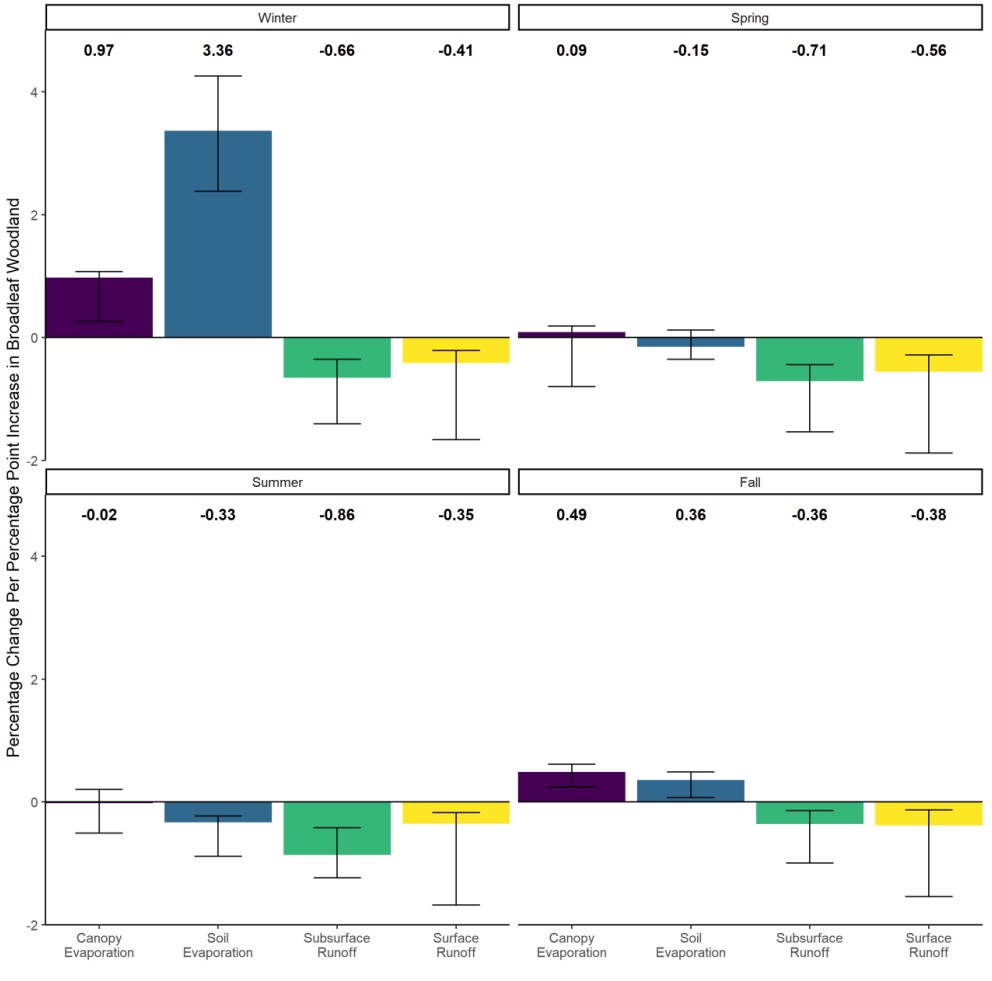

**Figure 4: Changes in the evaporative and runoff fluxes per percentage point of regional afforestation (PPPoA), e.g., 10% to 11% of a region afforested, by season with all other variables held constant. Error bars represent one standard deviation from the median value for the UKCP18 hydro-regions investigated.**



At lower streamflow quantiles the influence of afforestation location diminishes [Table 2]. At the top 1 % of flows there is no strong response to afforestation (-0.054 % PPPoA; $\rho$ = -0.2, p = 0.012), whereas the top 5 % of flows reduce by 0.11 % PPPoA ($\rho$ = -0.54, p < 0.01) with an even stronger median flow drop of -0.18 % PPPoA ($\rho$ = -0.65, p < 0.01) [Table 2]. At the lowest flow exceedances there are clearer patterns between streamflow reductions and afforestation with a decrease of -0.24 % PPPoA ($\rho$ = -0.66, p < 0.01) and -0.57 % PPPoA ($\rho$ = -0.81, p < 0.1) for the 90th and 99th flow

percentiles accordingly [Table 2]. We find an unclear picture of flow variability changes with afforestation extent, with a decrease 0.09 % PPPoA in the flow duration curve that appears more strongly related to differences in regional factors than afforestation itself ($\rho$ = -0.2, p = 0.012).

### 3.2 Hydrological Sensitivity to Climate and Land Cover Changes

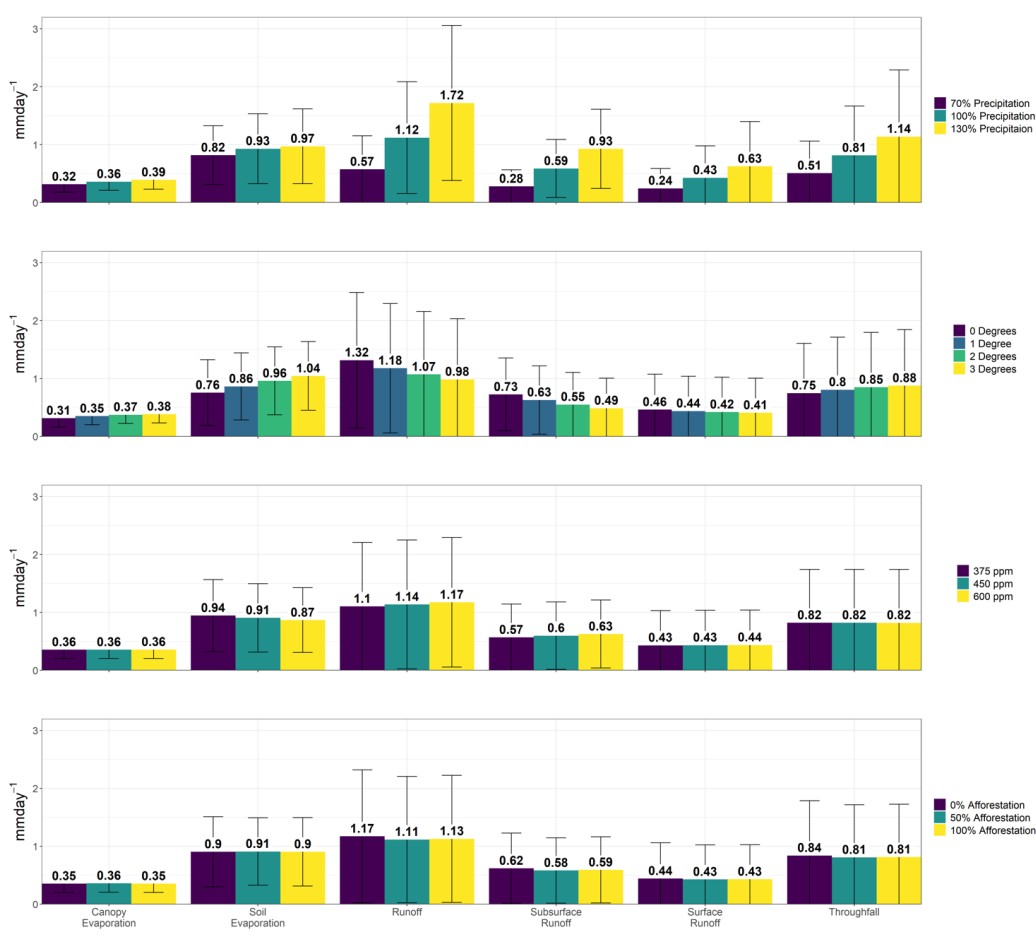


**Figure 5: Mean hydrological fluxes across all UKCP18 regions for each of the four variables altered relative to present climate and landcover: precipitation, temperature, $CO_2$ and landcover. Error bars indicate one standard deviation. Supplementary Figures S5 and S6 show how these change in summer and winter respectively.**





As the climate changes, land cover is also expected to change. Although it is expected that hydrological systems will respond significantly to climate change it is unknown what the relative response to concurrent land cover changes will be. Hydrological processes in JULES show strong sensitivity to climate relative to LULC across the range of scenarios tested [Figure 5]. ANOVA reveals significant differences in hydrological variables in all regions with proposed changes in precipitation and temperature ($p < 0.01$). When compared to projected changes in precipitation, temperature and $CO_2$,

the effects of LULC are almost undetectable. Only in a few isolated regions in winter are canopy storage, stomatal conductance and soil moisture significantly altered by LULC ($p < 0.01$) [Supplementary Material: Figures S3 & S4]. Interestingly, rising carbon dioxide only notably reduces transpiration and stomatal conductance in some regions (ANOVA, $p < 0.01$) [Supplementary Material: Figure S3]. Although not substantial, higher $CO_2$ suppresses soil evaporation, which increases soil moisture and therefore runoff.


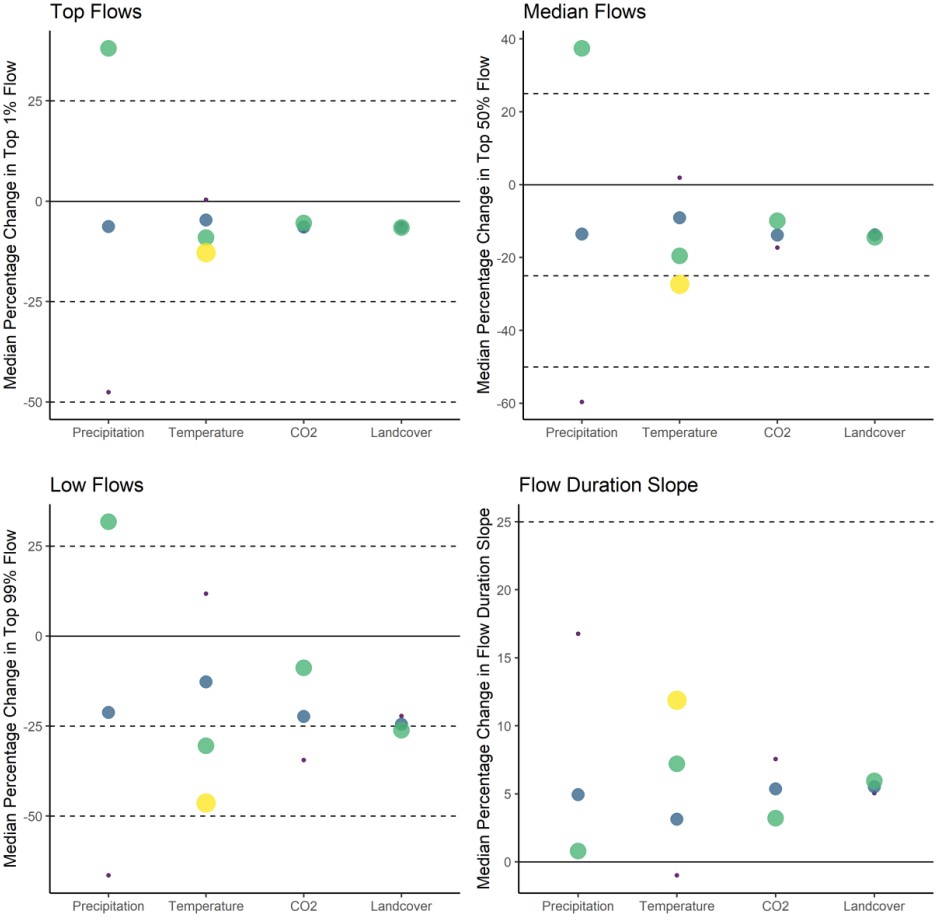

**Figure 6: Median percentage change in the indicated four metrics for catchments based on precipitation, temperature, CO2, and afforestation changes. Increasing dot size and lighter colour indicate larger changes in the variables explored (precipitation: 70%, 100%, 130%; temperature: 0 °C, +1 °C, +2 °C, +3 °C; CO2: 375 ppm, 425 ppm, 600 ppm; afforestation: 0%, 50%, 100%). Colours are used to further differentiate variable quantity and are not relative to the size of the variable. The solid black horizontal line indicates a 0 % change for all variables whereas the dashed lines indicate defined intervals of change.**






Enhanced precipitation across Great Britain greatly increases hydrological fluxes and stores compared to all other factors
altered [Figure 5]. When averaged across the whole period, precipitation does not have a significant impact on soil
evaporation and transpiration, except in northwest Scotland. However, in summer, enhanced precipitation significantly
increases soil evaporation across many regions (1.4 to 1.7 mm day$^{-1}$ from 70 % to 130 % precipitation) [Supplementary
Material: Figure S5]. Both canopy and soil moisture stores increase with precipitation, enhancing runoff (0.7 to 1.7 mm
day$^{-1}$ from 70 % to 130 % precipitation) [Figure 5].


Rising temperatures appreciably alter many of JULES' hydrological parameterisations. Soil and canopy evaporative
processes rise in winter with higher temperatures (both by 0.2 to 0.4 mm day$^{-1}$ with 3°C extra). In summer, soil evaporation
continues to increase (1.4 to 1.7 mm day$^{-1}$ with 3°C extra) but canopy evaporation stays approximately the same. This
slightly enlarges the canopy store, although not significantly for many regions. In winter, throughfall increases (0.8 to 1.2
mm day$^{-1}$ with 3°C extra) likely due to more intense rainfall which is parameterised by temperature. The higher throughfall
further reduces canopy storage. Stomatal conductance decreases throughout the entire period with rising temperatures but
is not significant for all regions and transpiration clearly increases with temperature in winter. Soil moisture reduces in
the summer with rising temperature which minimises subsurface runoff (0.4 to 0.2 mm day$^{-1}$ with 3°C extra). However,
there are no statistically significant changes in river flow and surface runoff in winter and summer for almost all regions
as temperatures rise.

Precipitation is a first order control on flood and drought formation in JULES, as expected. Precipitation significantly
(Kruskal-Wallis (KW) test, $p < 0.001$) influences the top 1 % of flows: reducing precipitation by 30% decreases them by
48 % whilst increasing precipitation by 30% enlarges them by 38 % [Figure 6]. Rising temperatures significantly (KW
test, $p < 0.001$) reduce high flows (+ 3°C reduces high flows by 13 %) [Figure 6]. Enhanced $CO_2$ insignificantly amplifies
the top flows whereas afforestation insignificantly reduces them [Figure 6]. Findings are similar at the lowest (top 99%)
flows, but the modelled flow response range is a greater with climate and land cover perturbations. Increasing precipitation
significantly increases low flows (KW test, $p < 0.001$) by 32 % for a 30 % precipitation increase (-67 % for a 30 %
precipitation decrease). Rising temperatures substantially reduce low flows by 46 % for an additional 3°C. Greater $CO_2$
increases low streamflow across all 108 scenarios (KW test, $p < 0.001$) from -34 % for 375 ppm of $CO_2$ to -8.8 % for 600
ppm. Across all the proposed environmental disturbances, LULC has the smallest impact on streamflow. Afforestation
only weakly decreases low flows from -22 % (0 % afforestation) to -26 % (100 % afforestation) insignificantly across all
scenarios in comparison to all scenarios (KW test, $p > 0.1$). Flow regimes became less variable with increasing
precipitation ($p < 0.001$, from 18 % to 0.60 %) and $CO_2$ ($p < 0.001$, reduction of 7.6 % to 3.2 %) [Figure 6]. In contrast,
rising temperature increases flow variability ($p < 0.001$) from -1.0 % to 12 % and afforestation increases flow variability
by only a small amount which is not statistically significant ($p > 0.1$).

### 3.3 Potential Influence of Afforestation in the Future

In the future, afforestation has a similar influence on hydrology as in the present climate [Table 1]. Therefore, projected
climate changes are insufficient to substantially alter simulated vegetation's interaction with water fluxes. Canopy
evaporation and storage are similarly influenced by land cover change location in the future ($\rho = 0.66$; $\rho = -0.94$
respectively) and increase by 0.33 % (0.47 mm yr$^{-1}$) and decrease by 0.64 % (0.001 mm yr$^{-1}$) PPPoA respectively [Table
1]. However, soil evaporation increases at half the present rate at 0.11 % (0.29 mm yr$^{-1}$) PPPoA and is strongly influenced
by afforestation location ($\rho = 0.51$). Transpiration robustly decreases at a rate of -0.78% PPPoA regardless of planting



location ($\rho$ = -0.98). Regional influence on stomatal conductance is increased compared to present ($\rho$ = 0.40) and rises
more rapidly at 0.14 % PPPoA.

Average simulated river flow, compared to present, drops at a slightly lower rate of -0.12 % PPPoA ($\rho$ = -0.82) [Table
1]. Runoff decreases with afforestation at a comparable rate to present (-0.27 %; 1.84 mm yr$^{-1}$ PPPoA) however, location
has a greater effect ($\rho$ = -0.74) and surface runoff (-0.16 %; -0.35 mm yr$^{-1}$) is greatly influenced by location ($\rho$ = -0.53)
[Table 1]. Subsurface runoff also becomes slightly more influenced by location ($\rho$ = -0.78) with a decrease of 0.26 % (-
0.95 mm yr$^{-1}$) PPPoA. Both throughfall and soil moisture respond to afforestation in a similar manner to as at present
regarding their trends and connection to planting location (-0.33 %, -1.23 mm yr$^{-1}$ PPPoA, $\rho$ = -0.83; -0.047 %, -0.41 mm
PPPoA, $\rho$ = -0.77 respectively) [Table 1].

In the future, planting location has a reduced influence on streamflow when compared to increases with afforestation
compared to present [Table 2]. Median streamflow reduces by -0.14 % PPPoA ($\rho$ = -0.68), low flows decrease by -0.25
% PPPoA ($\rho$ = -0.64) at the 90th percentile of flow and -0.72 % PPPoA ($\rho$ = -0.84) at the 99th percentile of flow [Table
2]. The effect of afforestation is more complicated at the highest flows with the top 1 % and 5 % of flows reducing by
0.11 % PPPoA ($\rho$ = -0.44 and $\rho$ = -0.66). Flow variability does not substantially change in the future, with insignificant
change (0.053 % PPPoA) and a strong regional influence ($\rho$ = -0.37).

## 4 Discussion

### 4.1 Afforestation Influence Across Great Britain

Regional hydrological response to afforestation does not significantly vary across Great Britain. Proposed countrywide
afforestation is projected with JULES to have a detectable, but not substantial, impact on hydrology. However, slight
nuances occur depending on afforestation location and the time of year (discussed later). LULC has a diminishing impact
on streamflow further from the intervention area, however the scale of afforestation considered here (large compared to
realistic afforestation rates) means modelled large-scale hydrological processes are detectable at multiple spatial scales
(Blöschl *et al.*, 2007; Pattison and Lane, 2012; Dadson *et al.*, 2017). The realism of the afforestation scenarios herein
illustrates how large-scale hydrological changes demonstrated by improbable widespread afforestation scenarios
considered in the literature, have minor relevance to the debate of probable afforestation rate impacts (Meier *et al.*, 2021;
Denissen *et al.*, 2022). Despite a broader modelling domain and greater hydrological diversity than in Buechel *et al.*
(2022), we find little difference in catchment water output sensitivity to afforestation location, suggesting potential
reductions in water yield can be directly estimated from the areal extent of woodland planted rather than its location.
However, this finding may simply demonstrate the relative insensitivity of terrestrial processes to landcover changes
modifying terrestrial processes within an LSM. No modelled hydrological change is exactly equivalent to afforestation,
for example a one percent woodland increase does not equal a one percent change in canopy evaporation. This is also
seen in observational studies where a 1 % increase in upstream afforestation area does not detectably change streamflow
(Anderson *et al.*, 2022).

The lack of regional variability in catchment hydrological response to afforestation could be due to terrestrial hydrological
similarity across the UK (Wagener *et al.*, 2021). Alternatively, the large epistemic uncertainty within JULES means that
highly sensitive hydrological parameters are not included that would lead to diverging regional afforestation responses,
such as variable vertical soil column properties (Beven and Cloke, 2012; Beven, 2018). LSMs scale physical processes



from very small areas and so the lack of sensitivity to widespread afforestation, and detected nuances related to location,
could also be due to inaccurate representations of model processes, such as the use of pedotransfer functions (Beven, 1989; Clark *et al.*, 2009). Another consideration is the uncertainty in soil products where there is large disagreement in the amount of organic material which significantly influences soil hydraulics (Feeney *et al.*, 2022). As an offline model (i.e., the land surface is not coupled to the atmosphere) factors that enhance regional importance with land parameterisations, such as orographic rainfall, are not represented. Importantly for this work, overall soil moisture dryness
may be overestimated as the rain seeding effect of afforestation is not included (Teuling *et al.*, 2017; Xu *et al.*, 2022). Modelling widespread afforestation with JULES has revealed further questions about the adequacy of LSM parameterisations.

The overall reduction in simulated runoff (and streamflow) by afforestation is a consequence of soil and canopy
evaporation increases [Figure 4]. Although evaporative processes are more influenced by regional properties [Table 1], JULES' land surface parameterisations screen atmospheric differences leading to diminished locational impacts on runoff compared to LULC. However, JULES systematically overestimates evaporation and so these results must be treated cautiously (Van den Hoof *et al.*, 2013; Blyth *et al.*, 2019). Rising evaporation rates are likely due to albedo reduction with afforestation, enhancing surface temperatures, and larger canopy stores with the higher LAI, and turbulence, compared
to grasslands. Increases in canopy storage, evaporation and interception reduce throughfall reaching the soil surface, which minimises regional climate differences relative to afforestation [Table 1]. Reduced throughfall means less soil moisture, runoff, and streamflow. The declining soil moisture with afforestation diminishes subsurface runoff and results in more water-stressed vegetation. The lower stomatal conductance, and thus transpiration, particularly in summer months, is evidence of the diminished soil moisture store. In JULES, broadleaf woodland has deeper roots than grasslands
and shrublands, which leads to water being extracted lower in the soil column to maintain growth (Best *et al.*, 2011; Harper *et al.*, 2021). In reality, tree root depths would be much deeper and vary according to the soil type (Vereecken *et al.*, 2022) and implementing woodland in this manner could lead to more accurate evaporation rates (Roebroek *et al.*, 2020; Harper *et al.*, 2021). In summer, moisture in the uppermost soil layer slightly increases which could be a function of lower roots, compared to grasslands, and reduced stomatal conductance (Buechel *et al.*, 2022). The slight association
of afforestation with topsoil moisture increases and location could be the result of different soil types (e.g., organic) which facilitate differences in hydraulic conductivity related to afforestation. The runoff model in this setup of JULES enhances runoff during high precipitation events with increased topsoil saturation, however, the proportionally small rise in topsoil moisture with simulated afforestation would make it potentially unobservable within natural uncertainty. Several limitations and assumptions should be considered when using LSMs such as JULES. The model domain only includes
known hydrological processes. Unincluded, or undiscovered, processes may have important consequences on afforestation's hydrological impact, unknown to the modeller (Beven *et al.*, 2011). Furthermore, processes within the model may inaccurately be implemented numerically or physically (Hrachowitz and Clark, 2017). Results therefore could potentially be affected by inadequate process representation and implementation.

Some regions in Great Britain exhibit slightly stronger effects of afforestation. In southeast England and Anglia, there are significantly larger hydrological variations, likely due to underlying soil properties and climate regime, similar to Buechel *et al.* (2022) with more sensitive catchments in drier regions. Afforestation therefore could strain water resources in regions of low water yield (Ellison *et al.*, 2012). However, the model representation of hydrological processes in groundwater-based catchments (found in these regions) is known to be inadequate (Le Vine *et al.*, 2016). Therefore in
reality, afforestation may have a muted influence on streamflow in these regions with roots accessing the deeper





groundwater (Roberts and Rosier, 2005). Evaporation rates are partially impacted by afforestation location, particularly in Scottish regions and parts of the west coast, which could enable flood magnitude reduction with spatially targeted broadleaf afforestation. The higher levels of wind turbulence and speed likely enable high evaporation rates to be maintained with additional woodland. Page et al. (2020) suggested canopy evaporation could reduce flood peaks in upland

regions; alignment between model and observations suggests further analysis to quantify evaporative processes over countrywide scales to mitigate flood risk. Regional differences in modelled stomatal conductance are due to climatic conditions, such as temperature and humidity as well as soil moisture and resulting vegetation water stress (Betts *et al.*, 2007; Best *et al.*, 2011). The similar connection of surface runoff with location suggests soil hydrology is an important control on stomatal conductance and runoff [Table 1]. JULES' modelling paradigm for simulating stomatal conductance

is not applied in all other models, and therefore other studies may find different projected stomatal conductance and thus resulting evaporation. However, JULES poses interesting questions for further exploration on the impact of widespread afforestation on regional hydrology and whether simulated changes are observable.

Afforestation across Great Britain influences the entire simulated streamflow spectrum (high to low flows) [Table 2]. It

is often observed significantly reducing the low to median flows, while high flow changes are frequently undetectable or inconsistent (Farley *et al.*, 2005; Do *et al.*, 2017; Anderson *et al.*, 2022). Afforestation's impact on streamflow is complex within JULES. Afforestation decreases the lowest flows and suggests locational factors have a minimal impact, similar to other research (Birkinshaw *et al.*, 2014; Bathurst *et al.*, 2020; Buechel *et al.*, 2022). If our projections are correct, water managers need to prepare for worse hydrological droughts with proposed afforestation in conjunction with those already

predicted in the future (Lowe *et al.*, 2018; Kay *et al.*, 2021). The mechanisms generating low flow response to afforestation are therefore more likely to be driven by runoff and soil moisture parameterisations compared to evaporative processes. This is because both runoff and soil moisture are more influenced by afforestation extent than location [Table 1] which is then replicated in the similar Spearman's rank correlation coefficients for the lower streamflow percentiles [Table 2]. To re-emphasise, larger Spearman's rank correlation coefficients indicate that afforestation extent, rather than locational

influences, impact the hydrological response. This highlights that the hydrological model structure within an LSM is likely to govern the model's ability to produce accurate drought predictions compared to other system parameterisations (Van Kempen *et al.*, 2021). In comparison, at the simulated very highest flows (top 1% of flows), afforestation both decreases and increases streamflow. This result is significant as more simple, and conceptually-based hydrological modelling often suggest afforestation reduces the highest flows (Stratford *et al.*, 2017). The regional differences in the

hydrological response of the highest flows to afforestation suggests evaporation rates are controlling the response, which is seen with the lower Spearman's rank correlation coefficients with afforestation extent and high flows, and evaporative processes. However, floods are often generated by extreme precipitation which generate 'numerical daemons' where the numerical implementation of hydrological processes generate implausibly high responses (Clark *et al.*, 2021; La Follette *et al.*, 2021). Therefore, floods could be more sensitive to terrestrial model parameters which could also lead to strong

regional influences on afforestation impact. Increases in small and large floods suggest similar generation mechanisms. Top flows usually decrease in catchments that are predominantly grass and pasture, where the chosen woodland planting criteria allow for larger areas to be afforested. Greater afforestation reduces simulated soil moisture, reduces throughfall and increases interception and canopy evaporation. When over five percentage points of a catchment's area is afforested, total soil and canopy evaporation rise, enhancing the catchment capacity to store and remove precipitation. During high

magnitude precipitation events, initial woodland planting reduces the overall effectiveness of the catchment to reduce flood peaks. Preliminary afforestation reduces the overall simulated maximal catchment water storage capacity in winter, due to decreased LAI, thus reducing water usage and evaporative fluxes.





These results have three major outcomes for future work. Firstly, models that encapsulate more known earth surface
dynamic processes (e.g., dynamic vegetation coupled with soil hydraulics and river runoff routines) produces a more
nuanced understanding of how afforestation could impact hydrology (Cooper *et al.*, 2021). Although not perfect,
particularly at hydrological extremes (Cuntz *et al.*, 2016; Brunner *et al.*, 2021), LSMs allow us to propose new hypotheses
related to the influence of afforestation (and LULC) on hydrological processes as a form of multiple working hypotheses
(Clark *et al.*, 2011). Secondly, analysis of median hydrological fluxes across hydro regions reveals reductions in overall
runoff and streamflow over an entire year which would lead to the incorrect assertion that afforestation could effectively
act as to mitigate peak flows over extensive areas as higher flows were less influenced. Targeted afforestation locations
within large catchments ($> 150$ km$^2$) may be ineffective for NFM downstream unless extensive or coupled with other
flood mitigation measures. Finally, model parameterisations have a significant bearing on our derived conclusions, with
hydrological model structure being more significant than other model parameters for determining afforestation impacts
on streamflow. Evaporative processes strongly influence simulated floods whereas runoff model implementations are
more important for calculating droughts with LULC change. Future work should therefore continue to investigate the role
of hydrological model structure within LSMs to assess its impact on quantifying the hydrological response to LULC
change (Clark *et al.*, 2021).

**4.2 Sensitivity to Climate and Afforestation Changes**

Compared to the most extreme proposed atmospheric changes, the impact of afforestation on Great Britain's hydrology
is relatively limited, with precipitation being the greatest driver of hydrological change compared to other variables
studied [Figure 5]. All water fluxes and stores rise in JULES with enhanced precipitation; in a projection of greatly
increased rainfall, flooding will likely increase, regardless of plausible land cover and other climate changes. For example,
with heavier rainfall, hotter temperatures and more $CO_2$ in winter (such as under the high-emissions Shared
Socioeconomic Pathway 5 (SSP5) – 'Taking the Highway' (Riahi *et al.*, 2017)) countrywide afforestation would not
reduce flood magnitude. Conversely, in summer, an overall decrease in precipitation (with increased temperatures and
$CO_2$) could greatly reduce runoff, which is only slightly diminished further by afforestation. In the UK, more intense
rainfall (convective) is predicted in the summer (Fosser *et al.*, 2020; Kay *et al.*, 2021; Kendon *et al.*, 2023) and our results
indicate realistic afforestation is likely to be ineffective for flood management during these events. The use in the present
study of JULES without atmospheric coupling, and no vegetation competition in this model configuration, means there
is no large-scale moisture recycling or vegetation mortality that would modulate model precipitation response further
(e.g. Cui *et al.*, 2022). For example, reduced rainfall could result in large-scale vegetation dieback, amplifying the effect
of high precipitation with reduced interception and less infiltration. Precipitation decreases demonstrate the nonlinear
parameterisations of JULES' hydrology with larger reductions at smaller streamflow percentiles [Figure 8] (further
justifying the factorial sensitivity analysis). Increased precipitation saturates the canopy and topsoil which quickly routes
excess water to rivers. There are two consequences of JULES' hydrological implementation. Firstly, uncertainty in
precipitation products measurably alters conclusions derived using JULES. Any small differences in precipitation used to
drive JULES will lead to larger differences in the modelled hydrological outputs. As a result, secondly, slight precipitation
product differences could negate the impact of LULC using JULES (or other similar LSMs) when comparing studies that
utilise different precipitation datasets. This is important as it suggests that work using LSMs to determine countrywide
changes in hydrology over periods where there have been relatively small land cover changes, can justify not using
evolving land cover as that would be likely to minimally reduce uncertainty (e.g. Blyth *et al.*, 2019). Further attention is


therefore required to minimise uncertainty in meteorological datasets to predict floods and droughts as terrestrial processes within LSMs will have a comparatively minor influence.

Previous work has illustrated the minor influence land cover has on hydrological processes compared to other atmospheric processes, both in models and in observational studies (Oudin *et al.*, 2008; Gedney *et al.*, 2014), and our results confirm that afforestation has the smallest impact on modelled streamflow compared to climate changes [Figure 5]. Therefore, to
detect afforestation influence upon streamflow, one must be aware of climate changes over the same period and be able to accurately remove any climatic effect which could obscure the LULC signal (Milly *et al.*, 2008; Slater *et al.*, 2021). This may explain why observational studies have found an insignificant impact of afforestation on streamflow (e.g., Anderson *et al.*, 2022), particularly with the large number of dependent interacting processes associated with woodland hydrology. Hydrological conclusions using JULES over long time periods therefore can determine that LULC is of minor
relevance when compared to climate (e.g., Blyth *et al.*, 2019). With climate change, afforestation is likely to be insufficient to reduce the largest pluvial flood risks. However, some research suggests smaller magnitude floods are becoming more frequent, and so plausible afforestation may mitigate the risk they pose (Griffin *et al.*, 2019; Wasko *et al.*, 2021). Our results emphasise LSMs pushing to be 'models of everywhere' are relatively insensitive to terrestrial process parameterisations in relation to climate drivers (Blair *et al.*, 2019). By applying atmospheric changes across the whole
country, variations in landcover, topography and soil type are insufficient to substantially alter the hydrological response. It is possible that in JULES, overparameterization could be leading to high complexity of interacting processes muting terrestrial parameter impact, or large epistemic uncertainty might be responsible for the minimal response compared to climate (Hrachowitz and Clark, 2017; Beven and Lane, 2022). Nonetheless, the JULES community model is continually being improved, and further work should test whether terrestrial properties, including LULC, are adequately represented.


Simulations of future climate suggest raised atmospheric $CO_2$ could negate the influence of increasing afforestation on streamflow [Figure 6]. With increases in $CO_2$, simulated streamflow rises across the flow spectrum because of reduced vegetation water usage. Amplified $CO_2$ decreases vegetation growth as the $CO_2$ pressure gradient between the stomata and atmosphere diminishes (Gedney *et al.*, 2006; Prudhomme *et al.*, 2014; Blyth *et al.*, 2019), which reduces soil water
usage, increasing soil moisture and overall runoff. JULES has exhibited strong sensitivity, be this correct or not, to $CO_2$ previously (Prudhomme *et al.*, 2014). If these results are accurate, afforestation in a changing climate may not be the silver bullet for mitigating flood risk and reducing atmospheric carbon policy makers envisage, particularly as vegetation becomes less effective at absorbing the additional $CO_2$ with increased atmospheric $CO_2$ (IPPC, 2019; Leung *et al.*, 2019; Cook-Patton *et al.*, 2020). A simpler hydrological model, and not an LSM, is unlikely to show the effect of $CO_2$ seen.
Atmospheric $CO_2$ has a strong control on low flows, which is important to consider in the context of future droughts and illustrates its strong influence on JULES' runoff mechanisms [Figure 6]. $CO_2$ fertilisation is not currently included within the version of JULES (vn5.6) used here, which would influence the effectiveness of vegetation to interact with water fluxes and potentially minimise the $CO_2$ impact (Bonan, 2008; Ritchie *et al.*, 2019; Zhang *et al.*, 2022). Therefore, afforestation may have an equivalent or larger impact on streamflow than $CO_2$, and we encourage further research to test
these results.

Temperature is a second order control on hydrological sensitivity within JULES and is more important than afforestation and $CO_2$ changes [Figure 5]. Streamflow is significantly reduced across the whole flow regime by increasing temperatures due to increased evaporation and water usage by vegetation. Again, looking under the high-emissions SSP5 scenario,
afforestation may enhance drought formation (both magnitude and duration) due to warmer temperatures. Increased flow





variability with higher temperatures is also likely to make it more difficult to adequately manage water resources. In JULES, precipitation is converted to convective rainfall at a certain temperature (the same amount of rainfall occurs in a smaller fraction of the grid box) (Best *et al.*, 2011). Temperature rises therefore trigger more 'convective' rainfall events but interestingly, even with increased precipitation, top flows do not grow, even with larger throughfall. This suggests

that in JULES, greater temperatures reduce antecedent soil moisture, decreasing runoff, and minimising the impact of the more intense rainfall. However, even the largest temperature increases cannot mitigate the impact of greater precipitation on increasing flood magnitude and frequency. Furthermore, this uncoupled model does not include realistic changes in rainfall intensity and magnitude which might be expected with increases in temperature and could change land surface responses to floods and droughts (Wasko *et al.*, 2019; Lee *et al.*, 2022).

**4.3 Afforestation Impact with Climate Change**

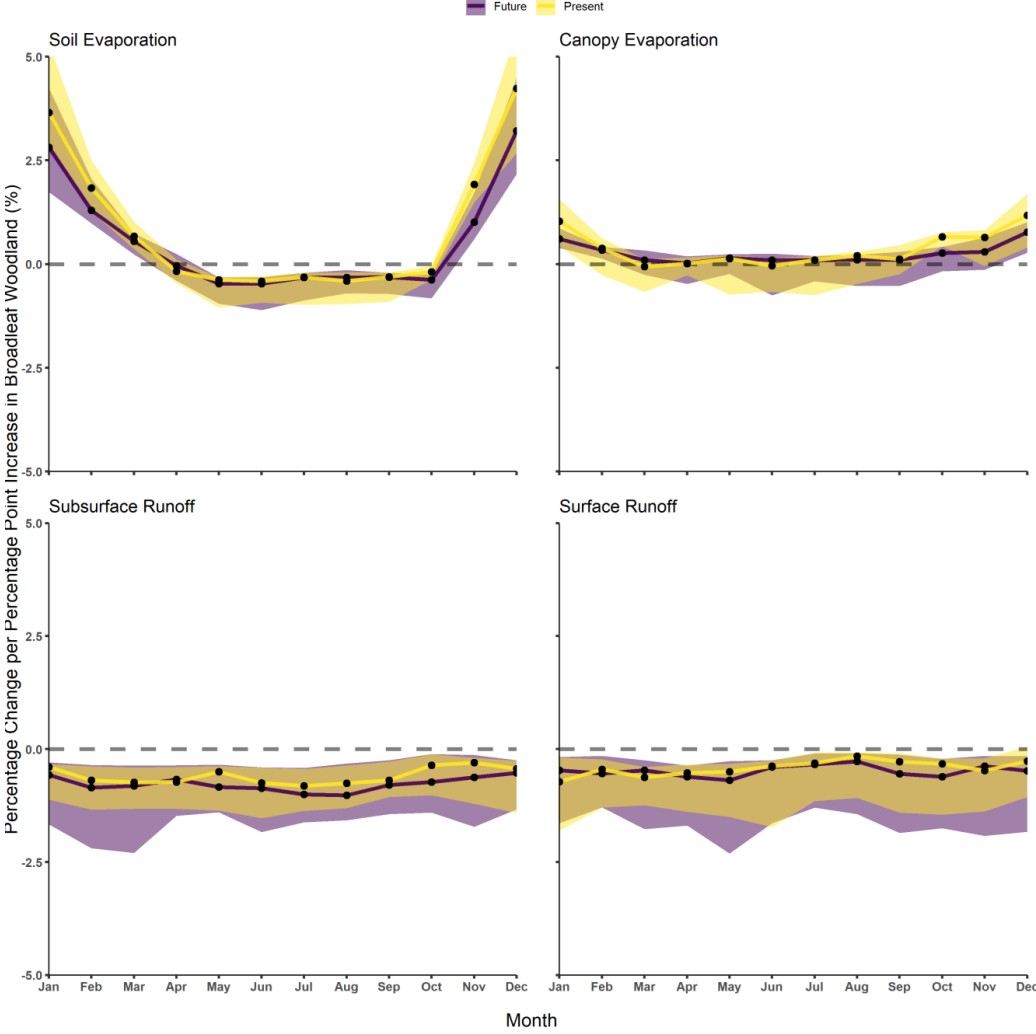

**Figure 7: Hydrological flux changes with PPPoA between the present (yellow) of 2000-2015 and future of 2020-2050 (purple). Use of an unpaired two-samples Wilcoxon test reveals no significant difference in the above hydrological fluxes between the**





**future and present (p > 0.1). Shaded regions represent the 10th and 90th percentile of hydrological flux changes across all**
**regions.**

In an extreme potential future climate scenario (RCP 8.5 / SSP 5), we find afforestation is unlikely to alter hydrological processes differently than under our present climate [Figure 7]. There are no statistically significant differences (KW test, $p > 0.5$) between the hydrological impacts for the same amount of afforestation in the present climate and the future [Figure 7]. Although future precipitation, temperature and $CO_2$ could have altered the woodland hydrological response,
as seen in the previous part of the study, the average climate changes are insufficient to induce significant changes. Afforestation is therefore unlikely to provide any greater protection from projected increases in hydrological extremes (Lane and Kay, 2021; Griffin *et al.*, 2022). This finding suggests that current estimates of the impact of afforestation on hydrology can also be used for the future; catchments where afforestation has reduced the largest floods are likely to continue to experience some protection for future events of similar magnitude. This is important as other work using
extreme land cover and climate scenarios suggest significantly different hydrological systems in the future. Projected climate changes are unlikely to be large enough to generate differing responses from land surface parameterisations in JULES, compared to the simple sensitivity analysis undertaken. Future studies should justify and utilise plausible land cover scenarios for policy recommendations to determine the future effect of changing climate and land cover more credibly on extremes when using numerical methods.


Some future impacts of afforestation on hydrological fluxes in JULES depart from present estimates. Transpiration decreases in the future, driven by lower summer precipitation, which mitigates the impact of growing broadleaf woodland relative to grasslands. Future rising temperatures, in conjunction with reduced albedo from more woodland, may therefore reduce stomatal conductance, due to the increased vapour pressure deficit. Decreases in both average river flow and runoff
appear to be more influenced by regional effects with a reduced correlation between amount of afforestation and percentage reduction in runoff across the studied regions. This is likely due to greater differences in precipitation and temperatures (previously shown as the main differentiator of streamflow response) leading to changes in evaporation and runoff. It might also suggest that evaporative processes have a stronger effect on runoff generation which have been shown to be driven by regional controls [Table 1]. However, a stronger correlation between the top 1 % of flows and
afforestation extent in the future [Table 2] suggests climate is likely to alter the modulating role of land cover during extreme events. Current implementations of land cover into LSMs need to ensure land cover parameterisations are accurate to ensure modelled responses to climate are faithful. If they are not, we are projecting further uncertainty into future scenarios.

**5 Conclusion**

Modelling 'realistic' countrywide afforestation in line with UK Government ambitions shows only small changes in hydrological processes and streamflow. Afforestation could generate unintended reductions in low flows in some locations, both at present and in the future. Although there are not significantly divergent regional responses to afforestation, catchment attributes and climate do produce nuanced hydrological responses (such as soil moisture). Evaporative processes govern high flow generation, while runoff parameterisation controls lower streamflow generation.
Our sensitivity analysis shows large-scale plausible afforestation has only a minimal impact on hydrology compared to possible climate changes. Precipitation changes have the largest impact on the modelled streamflow regime whereas temperature and $CO_2$ have a discernible impact on the lowest flows only. Furthermore, this study illustrates the epistemic uncertainties within the JULES model and potentially under-sensitive land surface parameters and parameterisations.





Finally, the effects of afforestation on land surface hydrology and the terrestrial hydrosphere are similar in the present
and future. Climate changes (e.g., precipitation and temperature) do not alter woodland regulation of hydrological
extremes and only slightly alter regional differences in the hydrological response to afforestation. Future research should
use fully coupled land surface – atmosphere LSMs to assess how afforestation influences hydrology over larger spatial
scales than the catchments studied to elucidate the strength and spatial extent of water cycling from increased canopy
evaporation.


*Code availability*. JULES' configuration details are found in Buechel et al. (2022) and accessible as Rose suite u-ce663
from the Met Office Rose/Cylc suite control system (https://metomi.github.io/rose/doc/html/index.html). The source code
for JULES can be found at: https://github.com/jules-lsm/jules-lsm.github.io.

*Data availability* NetCDF afforestation scenarios used to run experiments can be found at:
https://doi.org/10.5281/zenodo.7957084. The CHESS datasets used to run model experiments can be found at:
https://catalogue.ceh.ac.uk/documents/7de9790e-66a2-44b5-988e-283d764ef52f. CHESS-SCAPE can be found at:
https://dx.doi.org/10.5285/8194b416cbee482b89e0dfbe17c5786c. Additional data can be provided from the primary
author on request.


*Supplement.* Supplementary Material is provided with this paper.

*Author Contributions.* M.B. ran the model simulations, analysed the results, and wrote the paper. L.S. and S.D. provided
guidance, avenues of investigation and edited the paper.


*Acknowledgments.* We thank Toby Marthews and Emma Robinson for assistance with the original configuration of the
JULES model. The JASMIN CEDA service provided access to facilities and resources.

*Competing Interests.* L.S. is an editor for HESS. M.B. is funded by NERC (NE/L002612/1). L.S. is supported by UKRI
(MR/V022008/1) and NERC (NE/S015728/1). S.D. is funded by NERC (NE/S017380/1).





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
