# Peer review of "Afforestation impacts on terrestrial hydrology insignificant compared to climate change in Great Britain"

_Hydrology and Earth System Sciences, 2023_

## Author Comment (AC1)

**Response 1 to Reviewer of Manuscript: Afforestation impacts on terrestrial hydrology insignificant compared to climate change in Great Britain**

Comments/Text of the Reviewer is in **black**, our response is in **blue**.

It is important to understand the relative effects of afforestation and climate on the hydrology of river catchments. This manuscript considers the effects of both afforestation and climate change in detail for catchments in Great Britain and concludes that afforestation impacts are insignificant compared to climate change. The modelling work seems to have been carried out well and the manuscript is generally well written. However what I feel is missing is a comparison with measured data on the effect of afforestation, which is needed in order to check the results are realistic and that the conclusion is valid. The authors even state on L71 "LSMs should therefore quantify projected hydrological changes whilst modellers determine if outputs are realistic". My experience suggest the conclusions are valid but it is important to demonstrate that this true.

We thank the Reviewer for their kind comments on the modelling work and writing. We respond to the second part of this paragraph in response to the point below.

JULES LSM model simulations were carried out for 51 river catchments in the UK a range of different afforestation and climate scenarios. My understanding is that the baseline model was calibrated and validated against streamflow and soil moisture data. Then afforestation and climate simulations are carried out. The results show that "Afforestation across Great Britain moderately reduces average river flow by 0.17 % PPPoA over the year" where PPPoA is the per percentage point of afforestation. Is this value realistic? In my view all the results and conclusions rely on the model producing a realistic value here and there does not seem to have been any attempt to validate this value. I appreciate obtaining good data for this is difficult, but there are UK studies available. In the wetter parts of the country there are sites at Coalburn, Plynlimon, Balquhidder and in the drier parts at Blackwood, Alice Holt, Thetford Forest and Clipstone Forest. A lot of these consider a change from grassland to coniferous forests rather than broadleaf forests considered here. Also in many cases there are point scale measurements rather than changes in catchment river flows, but they will give an idea if the results are roughly correct. For example in the Coalburn catchment (Birkinshaw et al. 2014), which has already been cited, a change from grassland to a mature conifer forest (90% of the catchment) has produced a reduction in streamflow of around 350mm for an average river flow of around 900mm, which by my calculation reduces average river flow by 0.35 % PPPoA.

It could also be argued that there is no validation that the change in climate is also producing realistic results in the model. But I am not sure how this can be checked expect that the current validation against existing streamflow measurements covers "known floods and drought events".

In a revised manuscript we would substantially revise our description of the validation of the JULES model configuration used here, drawing on an extensive range of prior studies that have evaluated the model's performance in relation to both hydrology, soil hydraulics, land cover, and biogeochemistry. In this study we require a physically, and not catchment, based model to explore the hydrological response to changes in land cover and climate across the country. JULES' strength comes from the combination of processes and systems, and not just singular ones, thus enabling exploration into the interaction between systems (such as dynamic vegetation and hydrology). This does however make it practically impossible to validate every output, particularly as there is no such data at this spatial and temporal resolution across the country (a point to be explored further below).

Importantly, this work builds upon validation and the studies of others to undertake one of the most advanced studies to date in this field. The biomechanics of JULES has been continuously improved and utilised in this study. To note the work of Harper et al. (2016, 2021) that both validated and improved the plant functional types used in JULES and found that in temperate regions that JULES produced good agreement with latent energy fluxes. The initial implementation of plant hydrology mechanics within JULES can be found in Alton et al. (2009) and shows that global runoff agrees fairly well with this implementation as well as evapotranspiration. Van den Hoof et al. (2013) and Blyth et al. (2019) also illustrate that JULES gets the broad pattern of evapotranspiration changes correct. Martinez-de La Torre et al. (2019a) demonstrate further that land surface models with vegetation dynamics generate better drydown characteristics than large scale hydrological models.

Considering streamflow and runoff, the model configuration used is that developed by Martinez-de La Torre et al. (2019b) that achieve an NSE score of over 0.8 for the River Thames. Older versions of JULES have also been shown to get the broad structure of hydrological events and extremes correct (e.g. Prudhomme et al. 2011; Harding et al. 2014). More recent version implementations of JULES have produced even more accurate results (e.g. Lewis & Dadson 2021; Mathison et al. 2023).

The main model configuration used in this study can be found in Buechel et al. (2022) as stated on line 152. Buechel et al. (2022) provides a validation of streamflow, soil moisture and evaporation (including at Alice Holt) and these can be found in the methodology section of that study. At the broadleaf woodland sites there was both systematic over and underprediction of soil moisture with an overestimation of evaporation. However, the system responses were broadly correct.

The Reviewer hits the main scientific quandary: there is no way to validate something that has yet to occur (widespread afforestation across the UK). The studies cited by the Reviewer are all coniferous afforestation and at small scales (often less than < 5 km$^2$). Here, in contrast, the scenarios utilised are all broadleaf afforestation at countrywide/ large catchment scales. Processes that are relevant at small scales, such as forest management strategy and ditching, will not be explicitly represented within this form of model. Neither would it be expected that the model should include these processes, particularly as the observational studies do not necessarily agree on the direction of change (e.g. Stratford et al. 2017). JULES should not be treated as completely free of uncertainty , but it broadly produces the right result for the right reasons and most importantly, has a high degree

of fidelity. Precisely calibrating one output (such as streamflow) would compromise other processes within the model domain. This study provides a potential response of countrywide hydrology to afforestation.

We suggest adding a paragraph in the methodology section, providing details on validation studies that support the use of JULES to study land cover and hydrology changes.

Alton, P., Fisher, R.,  Los, S., and  Williams, M. (2009),Simulations of global evapotranspiration using semiempirical and mechanistic schemes of plant hydrology, _Global Biogeochem. Cycles, 23, GB4023, doi:[10.1029/2009GB003540](https://doi.org/10.1029/2009GB003540

Blyth, E.M., Martinez-de la Torre, A. and Robinson, E.L., 2019. Trends in evapotranspiration and its drivers in Great Britain: 1961 to 2015. Progress in Physical Geography: Earth and Environment, 43(5), pp.666-693.

Buechel, M., Slater, L. and Dadson, S., 2022. Hydrological impact of widespread afforestation in Great Britain using a large ensemble of modelled scenarios. Communications Earth & Environment, 3(1), p.6.

Harding, R.J., Weedon, G.P., van Lanen, H.A. and Clark, D.B., 2014. The future for global water assessment. Journal of Hydrology, 518, pp.186-193.

Harper, A. B., Cox, P. M., Friedlingstein, P., Wiltshire, A. J., Jones, C. D., Sitch, S., Mercado, L. M., Groenendijk, M., Robertson, E., Kattge, J., Bönisch, G., Atkin, O. K., Bahn, M., Cornelissen, J., Niinemets, Ü., Onipchenko, V., Peñuelas, J., Poorter, L., Reich, P. B., Soudzilovskaia, N. A., and Bodegom, P. V.: Improved representation of plant functional types and physiology in the Joint UK Land Environment Simulator (JULES v4.2) using plant trait information, Geosci. Model Dev., 9, 2415–2440, https://doi.org/10.5194/gmd-9-2415-2016, 2016.

Harper, A. B., Williams, K. E., McGuire, P. C., Duran Rojas, M. C., Hemming, D., Verhoef, A., Huntingford, C., Rowland, L., Marthews, T., Breder Eller, C., Mathison, C., Nobrega, R. L. B., Gedney, N., Vidale, P. L., Otu-Larbi, F., Pandey, D., Garrigues, S., Wright, A., Slevin, D., De Kauwe, M. G., Blyth, E., Ardö, J., Black, A., Bonal, D., Buchmann, N., Burban, B., Fuchs, K., de Grandcourt, A., Mammarella, I., Merbold, L., Montagnani, L., Nouvellon, Y., Restrepo-Coupe, N., and Wohlfahrt, G.: Improvement of modeling plant responses to low soil moisture in JULESvn4.9 and evaluation against flux tower measurements, Geosci. Model Dev., 14, 3269–3294, https://doi.org/10.5194/gmd-14-3269-2021, 2021.

Lewis, H.W. and Dadson, S.J., 2021. A regional coupled approach to water cycle prediction during winter 2013/14 in the United Kingdom. *Hydrological Processes*, *35*(12), p.e14438.

Martínez-de la Torre, A., Blyth, E. M., and Weedon, G. P.: Using observed river flow data to improve the hydrological functioning of the JULES land surface model (vn4.3) used for regional coupled modelling in Great Britain (UKC2), Geosci. Model Dev., 12, 765–784, https://doi.org/10.5194/gmd-12-765-2019, 2019a.

Martínez-de la Torre, A., Blyth, E.M. and Robinson, E.L., 2019b. Evaluation of drydown processes in global land surface and hydrological models using flux tower evapotranspiration. Water, 11(2), p.356.

Mathison, C., Burke, E., Hartley, A.J., Kelley, D.I., Burton, C., Robertson, E., Gedney, N., Williams, K., Wiltshire, A., Ellis, R.J. and Sellar, A.A., 2023. Description and evaluation of the JULES-ES set-up for ISIMIP2b. Geoscientific Model Development, 16(14), pp.4249-4264.

Prudhomme, C., Parry, S., Hannaford, J., Clark, D.B., Hagemann, S. and Voss, F., 2011. How well do large-scale models reproduce regional hydrological extremes in Europe?. Journal of Hydrometeorology, 12(6), pp.1181-1204.

Stratford, C., Miller, J., House, A., Old, G., Acreman, M., Duenas-Lopez, M.A., Nisbet, T., Burgess-Gamble, L., Chappell, N., Clarke, S. and Leeson, L., 2017. Do trees in UK-relevant river catchments influence fluvial flood peaks?: a systematic review. https://core.ac.uk/download/pdf/96704761.pdf

Van den Hoof, C., Vidale, P.L., Verhoef, A. and Vincke, C., 2013. Improved evaporative flux partitioning and carbon flux in the land surface model JULES: Impact on the simulation of land surface processes in temperate Europe. Agricultural and forest meteorology, 181, pp.108-124.

**Specific Comments**

L85. I note that in sections 3 and 4 that the sub-sections correspond to the 3 research questions. I might be worth highlighting this here as it does make the manuscript easier to read and I did not realise to start with.

We thank the Reviewer for this helpful comment, and we can amend the manuscript accordingly to change the subtitle headings to emphasise that they relate to the 3 research questions.

L161. "JULES runs at a numerical timestep of 30 minutes". This is clear, but what is the timestep of the meteorological input data and the streamflow data (I might have missed this)?

We are grateful to the Reviewer for highlighting the omission of this information. CHESS-met is given at a daily resolution and is disaggregated by JULES. The streamflow data calculated and compared to is at a daily resolution as well. This can be clarified within the manuscript.

L188. There should be a comparison with other hydrological models of UK catchments here, for example Lane et al. (2019) and Lees et al. (2021). These use NSE as an objective function rather than KGE, but I note in the supplementary material that NSE values are calculated. Also are you using hourly or daily discharge for the comparison?

We thank the Reviewer for raising this point and including a reference to our earlier work. Although it is beyond the scope to run more hydrological models to compare the model to, we will cite these references for readers to consult. Those studies also use KGE. It is heavily debated which calibration

factor is the most appropriate (e.g. Knoben et al. 2019). The daily discharge is used and can be clarified.

Knoben, W.J., Freer, J.E. and Woods, R.A., 2019. Inherent benchmark or not? Comparing Nash–Sutcliffe and Kling–Gupta efficiency scores. Hydrology and Earth System Sciences, 23(10), pp.4323-4331.

L199. Which Dee catchment? There are at least two rivers called Dee in the UK

We apologise for the confusion. In this context we are talking about the Dee UKCP18 river basin region, and this can be clarified within the text.

L217 "twenty UKCP18 river basin boundaries". Why are there 20 here but in L196 there were 51?

This is because this line refers to the 'UKCP18 river basin boundaries' whereas the 51 are for the catchments as written. We can emphasise this difference within the manuscript if required.

L233. "Proportional Influence of Afforestation Compared to Climate". You talk about changing precipitation, temperature, and CO2 (i.e three variables) but there are also changes in afforestation (so a fourth variable). I got confused and had to go back and read the section again.

We apologise for the confusion and will rewrite the section for clarity.

L288-292. This could do with being rewritten. The authors define soil evaporation "Simulated soil evaporation, including both evaporation from the soil surface and plant transpiration" but then consider transpiration separately but not soil surface evaporation separately.

These lines can be rewritten as there is no way to disentangle soil evaporation and soil transpiration. This line was written incorrectly and was meant to be modelled stomatal conductance and not transpiration.

L308 Table 1. How are "Soil moisture" and "Canopy Storage calculated". Is this the average over the simulation period? The data in the "future" columns is discussed in section 3.3, can the caption be changed to make this clear, when I first read this I got confused about why these results had suddenly appeared.

They are calculated in the same way the other variables are calculated. The table subtitle is now written 'Total Column Soil Moisture' to clarify that the soil moisture is for the whole column. The caption can be altered to explain for the references.

L312 Table 2. The correlations seems to be calculated for the percentage change. I would be interested in calculating the correlation for the absolute change. As I am sure the authors are aware many catchments in Scotland and North West England and Wales have precipitation totals that are around 3 times those in South East England. So if they have a similar percentage change they will have an absolute change that will be very different, which suggests there will be a significant influence due to the location for absolute change.

Flows across the catchments are different, as the Reviewer notes, however we need the changes to be on a comparative scale to observe the relative effect of woodland on flow percentiles. The use of a percentage scale enables us to deduce the relative impact of afforestation on hydrological processes across catchments of different sizes and hydrological magnitudes. Fundamentally the processes occurring in one area are the same elsewhere in the model domain and so use of the percentage change can be used to calculate the absolute change if needed (but could not be achieved the other way around). As shown in Table 2 and as written in the manuscript, location has a significant impact on the response of high river flow in comparison to woodland. If absolute values were used, the scale of the catchment and location would have a far greater impact on flow generation and not enable the influence of woodland to be ascertained (which is the question being asked here).

L325 Figure 4. In winter canopy evaporation increases as expected (Page et al. 2020). But what is driving the massive increase in soil evaporation. Soil evaporation "includes both evaporation from the soil surface and plant transpiration" but L291 says transpiration decreases with afforestation. This implies that increasing the forest is producing a massive increase in evaporation from the soil surface in winter. This does not make sense to me as a mature forest will have very little soil surface evaporation in winter. So either the model is doing something strange or I have misunderstood the results.

In winter there is diminished canopy protection due to the phenology implementation in the model, which leads to an increase in soil evaporation. On line 291, transpiration was the wrong word and was meant to be 'stomatal conductance' and will be changed in the manuscript. Further information on the evaporation effect can be found in Buechel et al. 2022 but it is due to the change from grasslands to woodland. To summarise a few points from that manuscript:

- Soil evaporation increased due to broadleaf woodland losing leaves. Reduced canopy cover increases soil exposure to incoming short-wave radiation and reduces aerodynamic resistance, which increases potential evaporation.

- In summer months a decrease in modelled soil evaporation increases topsoil moisture (because the increase in canopy foliage reduces soil exposure to short-wave radiation and surface wind speed).

- Topsoil moisture may increase because of root structure differences in broadleaf woodland and grasslands representation in JULES.

Lines 457-458 also explain that evaporation is likely to be overestimated in the model.

Buechel, M., Slater, L. and Dadson, S., 2022. Hydrological impact of widespread afforestation in Great Britain using a large ensemble of modelled scenarios. *Communications Earth & Environment*, *3*(1), p.6.

L340 Figure 5 In the bottom panel the runoff increases slightly and the canopy evaporation decreases slightly for 100% afforestation compared to 50% afforestation. The changes are small but I was wondering why this was happening.

This is likely due to means of the variables, runoff and canopy evaporation, being calculated and so the extreme increase in subsurface runoff being calculated for 100% afforestation is leading to that increase. For canopy evaporation, again there is such a spread that it is statistically insignificant when compared to other factors.

L360 Figure 6. I feel it would be easier to interpret if the y axis had the same scales for Top Flows, Median Flows and Low Flows

We understand the Reviewer's perspective on this and we initially presented the data with the same scale. However, it makes the data more difficult to interpret (particularly with the larger y axis range for the Median Flows). In Figure 6, we have used the same horizontal lines for comparison.

L406-L407 "Average simulated river flow, compared to present, drops at a slightly lower rate of -0.12 % PPPoA (ρ = -0.82) [Table1]. Runoff decreases with afforestation at a comparable rate to present (-0.27 %; 1.84 mm yr-1 PPPoA)". I do not remember seeing the difference between "river flow" and "runoff" defined. Can this be added somewhere and then explained why they are getting different results. Also Table 1 has runoff but no river flow. where is the value of -0.12% in Table 1?

Model definitions of river flow and runoff can be included. River flow is that taken at the gauging station location after the runoff has been routed through the kinematic wave River Flow Model, whereas runoff is taken as the specific combination of subsurface and surface runoff for the entire catchment.

River flow was removed from the table in editing and this figure reference can be removed. We thank the Reviewer for spotting this error.

L421 Discussion. The discussion is interesting but my personal feeling is that it is on the long side (I struggled to concentrate whilst reading it) and there are bits that are not completely relevant.

We are sorry that the Reviewer believes that there are irrelevant parts to the discussion. However, we have included everything we believe is necessary and this work is trying to remain of broad interest to those working on afforestation policy, woodland hydrology, and modellers in both hydrology and land surface models. It is therefore difficult to write a discussion that is relevant to just one group and thus the discussion is aimed at all those who might have an interest on this topic. However, we appreciate the Reviewer's point and we will attempt to streamline the text so that it does not feel unnecessarily long.

L432 "suggesting potential reductions in water yield can be directly estimated from the areal extent of woodland planted rather than its location". This follows from an earlier point this may be true for the percentage reductions but does the location affect the absolute reductions?

The trends in the absolute reductions are the same as the percentage reductions. Usage of percentage enables the changes to be on a relative scale between catchments and regions. Therefore, one can broadly convert the percentage changes into absolute changes.

L466 "In reality, tree root depths would be much deeper" is this due to using on a 3m deep soil column in JULES? Maybe make this clear.

We will edit the text to clarify, e.g. 'In reality, tree root depths would be much deeper than currently represented in JULES…'.

L485 "Therefore in reality, afforestation may have a muted influence on streamflow in these regions with roots accessing the deeper groundwater (Roberts and Rosier, 2005)." I do not understand this. If forests can access deeper groundwater then it might have a greater influence on streamflow in the longer term as it can transpire water even when there is a meteorological drought.

We understand the confusion of this and can replace 'muted' with 'more subtle and greater' to emphasise the fact of woodland impacting groundwater.

L513 "afforestation both decreases and increases streamflow". This needs explaining

We can add '… depending on catchment and antecedent conditions' to provide greater clarity.

L520-L527 Could this bit be removed? It is all a bit vague I do not really understand the point it is making.

We understand that the Reviewer may not be interested in this, but we believe it should remain to benefit the hydrological and land surface modelling communities. However, it can be edited for greater clarity if there is confusion.

L560 where is Figure 8?

We thank the Reviewer for finding this error from editing. The actual figure caption for this should have been Figure 6.

L580 "with climate…". Is this bit a repeat from the previous section?

This is not a repeat of the previous section. This is to highlight to the reader that if considering the role afforestation may have on flood processes, there is unlikely to be a significant impact of woodland being able to reduce these high flows.

L585 "By applying atmospheric changes across the whole country, variations in landcover, topography and soil type are insufficient to substantially alter the hydrological response." Has this been shown for topography and soil type?

Although there has been no explicit testing for soil and topography type, the alterations in hydrological processes are more substantial with atmospheric changes (e.g. precipitation) than due to the differences in land surface properties of the different regions. We have tested across a wide range of topographies and soil types with the number of catchments and find that the effect of land-cover on the response is much lower than the effect of changes in climate. Therefore, it can be concluded that within this model setup the driving atmospheric data is more important for generating hydrological responses. The lines 586-589 emphasise that this could be due to overparameterization in JULES or missing relevant processes.

**References**

Birkinshaw, S. J., Bathurst, J. C., & Robinson, M. (2014). 45 years of non-stationary hydrology over a forest plantation growth cycle, Coalburn catchment, Northern England. *Journal of Hydrology*, *519*, 559-573.

Lane, R. A., Coxon, G., Freer, J. E., Wagener, T., Johnes, P. J., Bloomfield, J. P., ... & Reaney, S. M. (2019). Benchmarking the predictive capability of hydrological models for river flow and flood peak predictions across over 1000 catchments in Great Britain. *Hydrology and Earth System Sciences*, *23*(10), 4011-4032.

Lees, T., Buechel, M., Anderson, B., Slater, L., Reece, S., Coxon, G., & Dadson, S. J. (2021). Benchmarking data-driven rainfall–runoff models in Great Britain: a comparison of long short-term memory (LSTM)-based models with four lumped conceptual models. *Hydrology and Earth System Sciences*, *25*(10), 5517-5534.

Page, T., Chappell, N. A., Beven, K. J., Hankin, B., & Kretzschmar, A. (2020). Assessing the significance of wet-canopy evaporation from forests during extreme rainfall events for flood mitigation in mountainous regions of the United Kingdom. *Hydrological Processes*, *34*(24), 4740-4754.

We would like to thank the Reviewer again for the time and effort they put into this review and we look forward to implementing their suggestions to improve the manuscript.

---

## Author Comment (AC2)

**Response 2 to Reviewer of Manuscript: Afforestation impacts on terrestrial hydrology insignificant compared to climate change in Great Britain**

Comments/Text of the Reviewer is in **black**, our response is in **blue**.

We thank the Reviewer for their time reviewing this article.

The study evaluates the potential hydrological changes under the possible afforestation regime alongside climate changes. Therefore, the results suggested that afforestation has an insignificant impact on the hydrology. The conclusion is not well supported by the modelling results since the overall model performance remains unsatisfactory. Model performance in most of the study sites was not satisfactory (in terms of NSE>0.5, which is more commonly used than KGE), which may require more detailed calibration and validation before future climate scenario is carried out. Further, some hydrography showing the changes might be useful for the presentation since the study evaluates hydrology.

In our response to the other reviewer, you can find a sample of validation studies that have shown the validity of using JULES to explore hydrological and land cover changes. The work here is ambitious in its scope by using a land surface model in which there are a multitude of processes from a range of systems beyond just hydrology. This makes it invaluable for determining complex questions such as that of woodland hydrology. For the ease of the Reviewer, we include a summary here of the key validation studies that illustrate the appropriateness of JULES in this study from our other reviewer response.

Harper et al. (2016, 2021) both validated and improved the plant functional types used in JULES and found that in temperate regions that JULES produced good agreement with latent energy fluxes. The initial implementation of plant hydrology mechanics within JULES can be found in Alton et al. (2009) and shows that global runoff agrees fairly well with this implementation as well as evapotranspiration. Van den Hoof et al. (2013) and Blyth et al. (2019) also illustrate that JULES gets the broad pattern of evapotranspiration changes correct.

Considering streamflow and runoff, the model configuration used is that developed by Martinez-de La Torre et al. (2019) that achieve an NSE score of over 0.8 for the River Thames. Older versions of JULES have also been shown to get the broad structure of hydrological events and extremes correct (e.g. Prudhomme et al. 2011; Harding et al. 2014). More recent version implementations of JULES have produced even more accurate results (e.g. Lewis & Dadson 2021; Mathison et al. 2023).

The main model configuration used in this study can be found in Buechel et al. (2022) as stated on line 152. Buechel et al. (2022) provides a validation of streamflow, soil moisture and evaporation (including at Alice Holt) and these can be found in the methodology section of that study. At the broadleaf woodland sites there was both systematic over and underprediction of soil moisture with an overestimation of evaporation. However, the system responses were broadly correct.

What the Reviewer is touching upon here is the idea of accuracy potentially above faithful representation of processes (Guse et al. 2021). Catchment hydrological models can be calibrated to produce accurate output (such as streamflow) but at the cost of compensatory parameters and an inability to elucidate realistic processes generating changes. This means that it is impossible to deduce potential realistic process responses to afforestation as a more finely calibrated model would alter processes to create an accurate output. We have chosen to use a physically based model that enables exploration of multiple systems and interactions between them.

As shown in the discussion we provide ideas for future improvement of JULES (e.g., lines 582-589) and acknowledge its limitations.

The review of Stratford et al. 2017 illustrates the issue of the current use of hydrological models to determine the impact of afforestation on flooding. Many models, although calibrated and validated to high accuracy, do not replicate the results seen in the observational data. As a community, we are therefore missing crucial processes in understanding the impact of woodland on hydrology within our hydrological models. We therefore need to turn to models of higher complexity that include more processes (especially phenology and stomatal conductance) to understand what the consequences of increased woodland area on hydrology could be. On lines 513-514 we highlight the fact that this model has generated more faithful and realistic results than compared to other models.

In relation to the KGE scores used, we would refer the Reviewer to Knoben et al. 2019 which highlighted that NSE and KGE are not comparable metrics. Although there is indeed a lot of work needed to improve hydrology in land surface models, they are satisfactory when we are more concerned about the multiple processes occurring and not just streamflow output. More validation information can be found in the papers highlighted on lines 150-154 in the main text. The same configuration used by Martinez-de la Torre et al. (2019) achieved NSE scores above 0.8 for the Thames catchment.

Guse, B., Fatichi, S., Gharari, S. and Melsen, L.A., 2021. Advancing process representation in hydrological models: Integrating new concepts, knowledge, and data. Water Resources Research, 57(11), p.e2021WR030661.

Stratford, C., Miller, J., House, A., Old, G., Acreman, M., Duenas-Lopez, M.A., Nisbet, T., Burgess-Gamble, L., Chappell, N., Clarke, S. and Leeson, L., 2017. Do trees in UK-relevant river catchments influence fluvial flood peaks?: a systematic review. https://core.ac.uk/download/pdf/96704761.pdf

Knoben, W.J., Freer, J.E. and Woods, R.A., 2019. Inherent benchmark or not? Comparing Nash–Sutcliffe and Kling–Gupta efficiency scores. Hydrology and Earth System Sciences, 23(10), pp.4323-4331.

Martínez-De La Torre, A., Blyth, E.M. and Weedon, G.P. (2019) 'Using observed river flow data to improve the hydrological functioning of the JULES land surface model (vn4.3) used for regional coupled modelling in Great Britain (UKC2)', 925 Geoscientific Model Development, 12(2), pp. 765–784. doi:10.5194/gmd-12-765-2019.

Alton, P., Fisher, R., Los, S., and Williams, M. (2009),Simulations of global evapotranspiration using semiempirical and mechanistic schemes of plant hydrology, _Global Biogeochem. Cycles, 23, GB4023, doi:[10.1029/2009GB003540](https://doi.org/10.1029/2009GB003540

Blyth, E.M., Martinez-de la Torre, A. and Robinson, E.L., 2019. Trends in evapotranspiration and its drivers in Great Britain: 1961 to 2015. Progress in Physical Geography: Earth and Environment, 43(5), pp.666-693.

Buechel, M., Slater, L. and Dadson, S., 2022. Hydrological impact of widespread afforestation in Great Britain using a large ensemble of modelled scenarios. Communications Earth & Environment, 3(1), p.6.

Harding, R.J., Weedon, G.P., van Lanen, H.A. and Clark, D.B., 2014. The future for global water assessment. Journal of Hydrology, 518, pp.186-193.

Harper, A. B., Cox, P. M., Friedlingstein, P., Wiltshire, A. J., Jones, C. D., Sitch, S., Mercado, L. M., Groenendijk, M., Robertson, E., Kattge, J., Bönisch, G., Atkin, O. K., Bahn, M., Cornelissen, J., Niinemets, Ü., Onipchenko, V., Peñuelas, J., Poorter, L., Reich, P. B., Soudzilovskaia, N. A., and Bodegom, P. V.: Improved representation of plant functional types and physiology in the Joint UK Land Environment Simulator (JULES v4.2) using plant trait information, Geosci. Model Dev., 9, 2415–2440, https://doi.org/10.5194/gmd-9-2415-2016, 2016.

Harper, A. B., Williams, K. E., McGuire, P. C., Duran Rojas, M. C., Hemming, D., Verhoef, A., Huntingford, C., Rowland, L., Marthews, T., Breder Eller, C., Mathison, C., Nobrega, R. L. B., Gedney, N., Vidale, P. L., Otu-Larbi, F., Pandey, D., Garrigues, S., Wright, A., Slevin, D., De Kauwe, M. G., Blyth, E., Ardö, J., Black, A., Bonal, D., Buchmann, N., Burban, B., Fuchs, K., de Grandcourt, A., Mammarella, I., Merbold, L., Montagnani, L., Nouvellon, Y., Restrepo-Coupe, N., and Wohlfahrt, G.: Improvement of modeling plant responses to low soil moisture in JULESvn4.9 and evaluation against flux tower measurements, Geosci. Model Dev., 14, 3269–3294, https://doi.org/10.5194/gmd-14-3269-2021, 2021.

Lewis, H.W. and Dadson, S.J., 2021. A regional coupled approach to water cycle prediction during winter 2013/14 in the United Kingdom. *Hydrological Processes*, *35*(12), p.e14438.

Mathison, C., Burke, E., Hartley, A.J., Kelley, D.I., Burton, C., Robertson, E., Gedney, N., Williams, K., Wiltshire, A., Ellis, R.J. and Sellar, A.A., 2023. Description and evaluation of the JULES-ES set-up for ISIMIP2b. Geoscientific Model Development, 16(14), pp.4249-4264.

Prudhomme, C., Parry, S., Hannaford, J., Clark, D.B., Hagemann, S. and Voss, F., 2011. How well do large-scale models reproduce regional hydrological extremes in Europe?. Journal of Hydrometeorology, 12(6), pp.1181-1204.

Van den Hoof, C., Vidale, P.L., Verhoef, A. and Vincke, C., 2013. Improved evaporative flux partitioning and carbon flux in the land surface model JULES: Impact on the simulation of land surface processes in temperate Europe. Agricultural and forest meteorology, 181, pp.108-124.

The conclusion is questionable based on the results of the study. For example, most of the changes in the broadleaf forest area are less than 10 per cent across the study sites. Although the scenario might be realistic under the conditions mentioned in the manuscript, however, not enough to support the conclusion since the hydrological change is not expected to be significant under the level of land cover change. Contradictory results are also referred to in the literature reviews, (L6: Many studies suggest afforestation can reduce overall streamflow). I suggest that results from the relevant studies should be discussed in the manuscript. Further, I believe that per percentage point of afforestation (PPPoA) is not a good indicator to evaluate the hydrological changes. I would suggest the model be simulated under selected paired-experiment sites (at least some examples should be carried out in the manuscript), which better considers the effects of afforestation.

The point being made here is not entirely clear to us. The magnitudes of afforestation are realistic, as stated on lines 132-134. The discussion of contrary findings in previous work provides important context. The choice of PPPoA metric allows for a consistent comparison across all locations. The suggestion to conduct paired-experiments is a good one but would be another study entirely. These points are elaborated below.

The question of this study was to understand the potential hydrological impact of widespread realistic afforestation in the UK compared to climate. As far as we are aware, there is no other study that has a conclusion to this question. Our results suggest that this potential land cover change (which is significant when compared to contemporary land cover changes) is insignificant when compared to potential climate changes.

The contradictory results discussed in the literature review emphasise that there is not a clear answer to the question of widespread afforestation on countrywide hydrology, hence the need for this study to provide further ideas and evidence.

We disagree that percentage point of afforestation is a bad indicator. This enables others to extrapolate the results over multiple spatial scales and see if they match our results, particularly as this work is trying to ascertain the relative impact of afforestation on hydrological processes.

The model used is not appropriate for the small spatial scales of paired-catchment studies ($< 10$ km$^2$) in which other more important processes (such as forest management) will have a more significant role on output (and are normally calibrated out and not understood by the modeller). We refer the Reviewer to lines 151-154 in which other studies have evaluated the streamflow, evaporation and soil moisture of this JULES configuration (as well as our response to the other reviewer). In addition, many paired-catchment studies in the UK with enough data to validate this model (as mentioned by

the other reviewer) are coniferous and this study does not consider this type of woodland expansion. However, we can add to the manuscript a few lines explaining why a validation study using a paired catchment approach would not be appropriate in this regard.

Various indicators are selected to evaluate the results, but most of them are not well defined in the methods sections or elsewhere in the manuscript (e.g. ANOVA/Kruskal). Terms such as high flow, low flow, summer, and winter should also be defined. The flow duration curve (FDC) is mentioned but rarely explained in the result section.

We thank the Reviewer for highlighting the need for clearer terms. The manuscript can be clarified accordingly. Flow percentile times and seasonal terms can be defined within the manuscript. The flow duration curve has been used in the results section (e.g. line 336 and Figure 6) and is useful for those seeking to understand flow regime change.

Some information about the model setup is missing. For example, it is unclear what temporal resolution is the results based on (hourly, daily or monthly), which is important for model performance evaluation. Also, the period of model simulations is not well described in the method section (e.g. L272 is unclear). The landcover data CAMELS-GB is not mentioned in the manuscript but is shown in the supplementary table.

Thank you for your feedback on this. CHESS-met is given at a daily resolution and is disaggregated by JULES to a 30 minute timestep. The streamflow data calculated and compared to is at a daily resolution as well. Please refer to the comments from the other reviewer in which we have clarified it is daily. The period of model simulations is stated on line 272 (2020-2050).

The supplementary table is meant to provide context for the reader of the various catchments. The data used to derive this data is the same as that used has been used to correct the driving data for JULES and we will clarify these aspects in the revised text.

**Specific Comments**

L107, L131 & L161 The land cover types are not consistent here, Non-default JULES land cover is referred to in L107 (acid grassland, arable and horticultural areas, heather, heather grassland, improved grassland and neutral grassland), however, how to turn it into 8 JULES land cover types are not mentioned.

The CEH Land Cover 2000 Map is used to identify the maximal possible extent of woodland (which is the same used to create CHESS-land base land cover). Broadleaf woodland is then planted as stated on lines 125-132. The land cover conversion can be found in the supplementary material of Buechel et al. 2022 and the manuscript can be rewritten to alert readers to this fact.

Buechel, M., Slater, L. and Dadson, S., 2022. Hydrological impact of widespread afforestation in Great Britain using a large ensemble of modelled scenarios. *Communications Earth & Environment*, *3*(1), p.6.

L128 I assume the percentage of changes was compared between the CEH 2000 landcover map (the year 2000), and scenarios in the year 2050 (but the selection of dataset was not well described, also the year 2050 is not pointed out clearly).

The manuscript can be rewritten to clarify that the change in hydrological processes for the three parts of this study is between the created afforestation scenarios and the CEH 2000 landcover map.

L159 temporal resolution of CHESS-met is not mentioned in the manuscript, which should be 1 hour.

The Reviewer is incorrect; the temporal resolution of CHESS-met is daily.

L162 Broadleaf tree (and other PFTs) setup: what is the parameter set used in this study? I would believe default parameters are used, however, should be mentioned somewhere in the manuscript.

The configuration is mentioned in the studies referenced on lines 151-154 and can be found in the Rose suite mentioned in the code availability section.

L183 The Kling-Gupta Efficiency Measure (KGE) formula should be described here if it is used for assessment. NSE (also found in the supplementary document) should be a more common indicator for hydrological evaluation.

We have included the formula for KGE in the supplementary text. We respect the Reviewer's opinion that 'NSE … should be a more common indicator' but that is a debated opinion (e.g. Knoben et al. 2019).

Knoben, W.J., Freer, J.E. and Woods, R.A., 2019. Inherent benchmark or not? Comparing Nash–Sutcliffe and Kling–Gupta efficiency scores. Hydrology and Earth System Sciences, 23(10), pp.4323-4331.

L186 This should be the results rather than the method. Further, the temporal resolution (hourly, daily or monthly?) of the results is not defined.

We disagree that the validation metrics are results, as the study is exploring the influence of afforestation on streamflow and is not a validation study.

L196 The location of 51 study sites should be pointed out here or in the supplementary document.

The manuscript already refers the reader to Supplementary Material: Figure S1 & Table S4 on Line 196 which is a map and description of the catchments.

L198 Abbreviation as UKBN2 should be explained when it is first referred to in the manuscript.

This can be included in the manuscript.

L202 "that processes simulated are more faithful at JULES' spatial (1 km2) and temporal (30 minutes)" the statement needs more explanation.

A sentence can be added afterwards to explain further: *This is because land surface models are intended to explore processes at countrywide and continental scales.*

L219 Season should be defined here although I believe it follows spring MAM, summer JJA …

Thank you for this recommendation which can be added to the manuscript.

L220 & L227 Theil-Sen slope estimator and Spearman's rank correlation coefficient formula should be explained if it is relevant for evaluating the results.

L224-L232 explains the use of these metrics and further reading regarding Theil-Sen.

L246 ten years (using 2000-2001): should be a typo here.

Data from the period 2000-2001 were used ten times to get the model spun-up. We will clarify this point in a revised manuscript.

L264 To clarify, The CHESS-SCAPE dataset (RCP 8.5) should be referred to in the first place.

In the structure of the manuscript, the third part of the study uses CHESS-SCAPE and thus it is referred to in the last part of the methods.

Table 2 Using percentages to describe the change of FDC might be not that meaningful.

As we are trying to quantify the change in the flow regime, we believe this is appropriate, despite not giving a significant result.

L345 "As the climate changes, land cover is also expected to change." It seems not to change in the three scenarios used in this study.

L345-349 places this comment in context. This study is focusing on the hydrological effects of afforestation in comparison to climate change. We are therefore comparing land cover change, in the form of woodland cover, to potential climate changes when using a land surface model. This work is not proposing overall changes in land cover as it is impossible to predict the exact evolution of land cover change in conjunction with climate and would require a whole separate study.  This sentence alone should not be taken as a description of the model experiment, but rather a statement of how the Earth is expected to change. We can clarify the text accordingly.

L348 ANOVA is not explained in the manuscript.

We will clarify that ANOVA stands for Analysis of Variance and provide a reference to the relevant statistical texts to define.

L371 "Rising temperatures appreciably alter many of JULES' hydrological parameterisations." Should affect JULES hydrological processes, not parameterisation.

We can alter the manuscript accordingly.

L383 Kruskal-Wallis (KW) test is not explained in the manuscript.

This statistical test can be described further in the manuscript.

L385 High flow/low flow are not defined in the manuscript.

Thank you for bringing this to our attention and will be clarified in the manuscript.

L437 "This is also seen in observational studies where a 1 % increase in upstream afforestation area does not detectably change streamflow" This could not conclude that" Afforestation impacts on terrestrial hydrology are insignificant."

We would disagree with the Reviewer on this point. This form of increase in afforestation is realistic in many areas and therefore, objectively, a 1% increase not detectably altering flow is hydrologically insignificant when compared to other factors (such as climate).

L441 "Alternatively, the large epistemic uncertainty within JULES means that highly sensitive hydrological parameters are not included that would lead to diverging regional afforestation responses." However, this part should be important to evaluate the hydrological regime.

We agree and that is why L440-452 raise these important points to the modelling community.

L560 Figure 8 is missing.

Thank you for noticing this. It is meant to refer to Figure 6 and can be altered accordingly.

L585 "By applying atmospheric changes across the whole country, variations in landcover, topography and soil type are insufficient to substantially alter the hydrological response" This is not real, soil parameters could considerably affect the hydrological regime, and it is not changed in this study before coming to this conclusion.

Soil parameters vary across catchments and regions of the UK. In the land driving data, these parameters were not enough to vary the responses generated by changes in climate and so we disagree with the Reviewer's point. The following lines of 586-589 put this comment in context.

L666 "Future research should use fully coupled land surface–atmosphere LSMs" I disagree that JULES has already reached its limitation here, there is more room to be improved for better model performance.

We agree that there is further model improvement (as was raised throughout the discussion) to be had. Therefore, we can rephrase that it 'could' rather than 'should', be potential further research.

We would like to thank the Reviewer for their time and helpful suggestions, which will help us to improve the manuscript.

---

## Author Response (AR1)

**Afforestation impacts on terrestrial hydrology insignificant compared to climate change in Great Britain**

**Response to the Editor**

Comments/Text of the Editor is in **black**, our response is in **blue.**

We thank the Editor for providing the opportunity to revise the manuscript and to provide responses to their helpful comments.

Dear authors,

Thank you for responding to the two reviews. The reviewers' comments are mixed, and contain some reservations. Your revised manuscript will be sent to both reviewers again. In addition to their comments, I have additional comments listed as follows.

1. The title may be misleading as you only assessed broadleaf woodland, I suggest you add broadleaf to the title. This also leads to my next question.

We will change the title of the manuscript accordingly.

2. You did not explain why only broadleaf woodland is investigated in your study. Please see below excerpt from https://cdn.forestresearch.gov.uk/2022/10/UKFSPG027.pdf that points out conifers have much great water use compared with broadleaves, and differences in effects tend to be greater between broadleaf species.

"Tree type has a marked effect on water use and thus the impact on flood volumes. Water use is generally much greater for conifers compared with broadleaves, especially in winter periods. Conifer canopy interception is typically twice that of broadleaves, resulting in larger evaporation losses during storm events and higher and more sustained soil moisture deficits, with a greater capacity for conifer soils to absorb rainwater and reduce flood run-off (but also to increase water shortages in drought periods). By contrast, soil infiltration rates tend to be similarly high under both tree types, provided woodland is well managed (e.g. avoiding ground compaction), while the hydraulic roughness created by trees and associated shrubs, ground vegetation and deadwood tends to be significantly greater under broadleaves, depending on tree age and the depth of floodwaters.

Tree species effects are generally small, especially within conifers. Differences tend to be greater between broadleaves, with the relatively high water use of willow and poplar species (when well supplied with water, such as in riparian and floodplain habitats) resulting in higher soil water deficits and a greater potential for below-ground floodwater storage. Tree species selection should be primarily driven by site suitability and habitat, but water impacts should be considered where significant."

We thank the Editor for noting that the use of just broadleaf woodland is not clearly explained in the original manuscript. This has now been edited to ensure that it is properly explained. Only broadleaf is used as a representative vegetation for woodland across the UK for several reasons. Firstly, the implementation of broadleaf in JULES is shown to be more accurate than needleleaf (Broadmeadow *et al.*, 2018). Secondly, at the spatial and temporal resolution of the model it would be inaccurate to extrapolate precise species responses to these scales. Finally, there is no clear map of the precise plant species that could be planted in different locations across the country, therefore this work aims to provide an estimate of the potential hydrological responses to afforestation, from which others can draw inferences about how individual tree species may compare.

Broadmeadow, S. et al. (2018) Valuing flood regulation services of existing forest cover to inform natural capital accounts, The Research Agency of the Forestry Commission. Available at:
https://www.forestresearch.gov.uk/documents/5499/Final_report_valuing_flood_regulatio n_services_051218.pdf.

3. The Plausible Afforestation Scenarios you used need further justification.
I checked the Forest Research, 2021a cited by you. They state '13.3 thousand hectares of new woodland were created in the UK in 2020- 2021, with conifers accounting for 55% of this area.'
The two scenarios you proposed are (1) 900 000 hectares of broadleaf woodland is randomly 'planted', at a 25 m resolution (2) 450 000 hectares of woodland is made to represent afforestation at similar present rates (Forest Research, 2021a). Present rate of 13,300 is not similar to 450,000.

You also stated "Arguably these scenarios are a restrictive level of afforestation, but they appear ambitious when compared to current afforestation rates of approximately 10 000 hectares per year (Forest Research, 2021b)." This is a vague statement, and did not explain why your scenarios are 90 or 45 times of the current rate. Is it because the changes would be negligible if not ambitious? It is also not clear if the 900k hectares broadleaf woodland is newly planted or includes the existing one. According to (Forest Research, 2021a), the area of woodland in the UK at 31 March 2021 is estimated to be 3.2 million hectares. If we take away 9% in NI, the area for conifers is 1,601,600 and remaining is 1,310,400 hectares. I still cannot see where 900k or 450k comes from.

Thank you for noting that this point requires clarification. It is correct that the amount of woodland planting is approximately 10 000 hectares per year in the UK. The government seeks to increase this rate to 30 000 hectares per year as specified in the manuscript (Committee on Climate Change, 2018). The 900 000 hectares of additional woodland is equivalent to 30 000 hectares over 30 years (this is now clarified in the manuscript). The 450 000 hectares is equivalent to 15 000 hectares over 30 years. This rate is higher than the 10 000 hectares currently being planted and that is why we say that rate would be ambitious.

We would also like to emphasize that this work is for Great Britain and thus does not include Northern Ireland (NI).

Committee on Climate Change (2018) 'Land use: Reducing emissions and preparing for climate change', (November), p. 100. Available at: www.theccc.org.uk/publications.

4. Figure 2: numbers at 50% and 100% don't seem to match up even if rounding errors are taken into consideration. For example, 5.6 and 11.4.

This is because the weighting mechanism for planting trees in our conceptual planting model means it is not necessarily spatially uniform across the country when randomised.

5. In your response to Reviewer 2, you state the follows.
"Catchment hydrological models can be calibrated to produce accurate output (such as streamflow) but at the cost of compensatory parameters and an inability to elucidate realistic processes generating changes. This means that it is impossible to deduce potential realistic process responses to afforestation as a more finely calibrated model would alter processes to create an accurate output. We have chosen to use a physically based model that enables exploration of multiple systems and interactions between them."
It is possible for an HM to deduce potential realistic process responses to afforestation. Please note there are physically based hydrological models that "enables exploration of multiple systems and interactions between them". In most cases, HMs are used for simulating streamflow and hence calibration centres on streamflow. They can also be calibrated using multiple criteria where not only streamflow can be calibrated but also other intermediate outputs within a HM.

We agree with this statement in the broadest terms but stand by the point made in our response. It is not entirely clear what the precise definition of a HM is in this case. Our statement is still accurate as HMs can generate 'realistic process responses to afforestation' but this can come from compensatory parameters or high epistemic uncertainty by not including faithfully all relevant processes. There are a range of hydrological models covering a spectrum of configurations and purposes; we are not saying that the model used here is better than other hydrological models, but that it is more appropriate for the questions asked in this manuscript.

6. You stated "Considering streamflow and runoff, the model configuration used is that developed by Martinez-de La Torre et al. (2019) that achieve an NSE score of over 0.8 for the River Thames." Your study covers GB. What are the NSEs for all the catchments covered in your study.

These are provided in the supplementary material (stated in the main manuscript).

7. Both reviewers asked for comparison with experiment sites. You say "The question of this study was to understand the potential hydrological impact of widespread realistic afforestation in the UK compared to climate." But if your results cannot be validated against several experiment sites in GB, how much credibility is it in understanding the potential hydrological impact of widespread realistic afforestation? What is the spatial resolution of your model setup? Did you use the Hydro-JULES? "LSMs are intended to explore processes at countrywide and continental scales". Is this why you don't validate your results using the experiment sites?

A new paragraph is added on the validation studies undertaken with JULES. The spatial resolution is stated explicitly in the manuscript in the first paragraph of Streamflow Analysis subsection of the Methods. We used JULES; Hydro-JULES is the name of a project to improve the hydrological representation within JULES. As previously stated, the reason why results cannot be validated for our specific experiment is that there are no experiments at this scale. Thus, we need to use models that can extrapolate the potential consequences of afforestation using physically represented processes.

8. Reviewer 2 commented "Further, some hydrography showing the changes might be useful for the presentation since the study evaluates hydrology." Can you please respond to this comment?

We apologise for not answering this comment before. We respect the point the Editor and the Reviewer make; however, this is already a complex and long piece of work and we believe that adding more figures will not add to the story. In particular, considering the spatial and temporal scales of the study, we feel that adding a hydrograph of a single catchment would not be particularly helpful to address the questions asked in this work.

I look forward to receiving your responses and the revised manuscript.

Sincerely,
Yi He, HESS Editor

We are grateful for the Editor's questions, which have helped us clarify important points in the manuscript.